# Multi-Task Sequence Models Generalise in Offline Multi-Agent Reinforcement Learning

## Abstract

Recent sequence model architectures have demonstrated great promise in offline multi-agent reinforcement learning (MARL). However, even for this expressive model class, generalising to tasks unseen in the training data remains a core challenge. A sensible response to this challenge is to simply scale the amount of offline data available for training. Yet, in this work, we find that task diversity has a stronger influence on generalisation than sheer dataset size. To obtain our findings, we study offline MARL sequence models trained on single-task datasets, clearly demonstrating their limited ability to zero-shot transfer to held-out test tasks. Leveraging this insight, we train and test multi-task versions of offline sequence modeling architectures. We identify three key design choices for successful offline multi-task training: (i) task-balanced mini-batches, (ii) treating value estimation as classification and (iii) agent masking to handle variable team sizes. Using large multi-task datasets from three challenging cooperative environments (`Connector`, `RWARE`, and `LBF`), we investigate generalisation to unseen tasks and the scaling behaviour of our multi-task offline algorithms. **We show that our multi-task sequence models generalise better across all environments compared to single-task models, and achieve a mean improvement of approximately 3.2x on held-out test tasks.** Moreover, our offline MARL sequence models consistently outperform behaviour cloning (a surprisingly strong baseline). Our results clearly show that scaling task diversity by increasing the number of tasks used during training leads to improved generalisation gains over simply scaling the dataset size at a fixed level of task diversity.

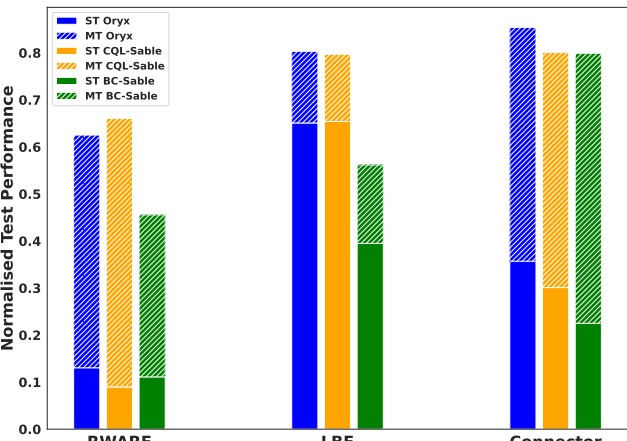

Figure 1: *Test task performance difference between single-task and multi-task sequence models.* Three multi-agent sequence models—CQL-Sable, BC-Sable and Oryx (Formanek et al., 2025)— were trained using either a single task (ST) or a set of multiple training tasks (MT). Average zero-shot performance was measured across a held-out set of test tasks. The upper bar represents the performance gap between ST and MT sequence models on unseen test tasks. **Averaged across all three algorithms, we observe a test performance increase of approximately 5.4x on `RWARE`, 1.3x on `LBF`, and 2.9x on `Connector`.**

# 1 INTRODUCTION

Building agents that generalise to tasks beyond those present in their training data is a central challenge in reinforcement learning (RL), and a prerequisite for deploying agents in the real world (Kirk et al., 2023). In many domains, collecting fresh data online by interacting with a live system is costly or risky, so practitioners turn to offline RL from logged trajectories (Levine et al., 2020). While single-agent work has studied the train–test generalisation gap (Mediratta et al., 2024), the multi-agent case remains under-explored. Despite recent progress in offline multi-agent reinforcement learning (MARL) (Yang et al., 2021b; Shao et al., 2023; Meng et al., 2023; Li et al., 2025; Formanek et al., 2025), prior work have largely been restricted to training and evaluating on the same task, without examining generalisation to unseen tasks.

In this work, we study the generalisation of single-task models, and then introduce a challenging multi-task benchmark for offline MARL, which builds on widely adopted MARL environments Level Based Foraging (LBF), Multi-Robot Warehouse (RWARE) (Papoudakis et al., 2021), and Connector (Bonnet et al., 2024). Using this benchmark, we evaluate three state-of-the-art offline multi-agent sequence models, namely Oryx (Formanek et al., 2025), as well as two offline versions of Sable (Mahjoub et al., 2025) (CQL-Sable and BC-Sable). Across all three environments, we show that these models exhibit poor generalisation when trained only on a dataset from a single task. However, when trained *simultaneously* on a dataset consisting of a diverse set of multiple tasks, their ability to zero-shot transfer to unseen tasks significantly improves. Furthermore, we verify that similar results cannot be obtained by simply increasing the size of the dataset for a fixed number of tasks, but rather that the key driver is increasing dataset diversity by adding more tasks, which consistently leads to improved test performance. Finally, we find that for a fixed data budget, increasing the model's capacity has a positive impact on generalisation for challenging tasks.

We identify three key design choices for multi-agent sequence models to be successfully trained across multiple tasks simultaneously: (i) task balanced batching, which makes the model unbiased over a mixture of tasks, (ii) value learning via classification (Farebrother et al., 2024) which improves the models ability to handle tasks with varying reward scales (Kumar et al., 2022a), and (iii) masking and shuffling active agents in the sequence, which allows the models to dynamically handle varying numbers of agents across tasks.

Our findings show that offline MARL sequence models trained on diverse multi-task datasets show promising signs of generalisation to unseen tasks, as compared to single-task alternatives. In contrast to the findings of Mediratta et al. (2024), we observe that our offline MARL methods do outperform behaviour cloning, a consistent and surprisingly strong baseline to beat. Finally, our work discovers the first promising signs of performance scaling (Hilton et al., 2023) with increases in model capacity for offline MARL on difficult unseen tasks.

In summary, our main contributions are as follows:

- We develop a challenging multi-task offline MARL ~~benchmark~~evaluation suite, which includes 30 large training sets and 22 test sets across LBF, Connector, and RWARE.

- We present two ~~novel~~ MARL sequence models (BC-Sable and CQL-Sable) that build on Sable (Mahjoub et al., 2025) by incorporating two widely used offline losses BC and CQL. We also validate three design choices that enable these models — and Oryx (Formanek et al., 2025) — to be trained on multi-task datasets.

- We show that the zero-shot generalisation capacity of all three multi-agent sequence models scales significantly (3.2x on average) as the number of tasks in the training data increases.

- We study the effect of dataset and model size on generalisation, clearly establishing that sheer dataset size in not the main driver of test performance, and that for difficult tasks, model scaling positively affects generalisation.

- **All of our (anonymized) code is available for download[1]. We will make all of our code and datasets publicly available upon publication.**

---

[1] https://sites.google.com/view/multi-task-marl

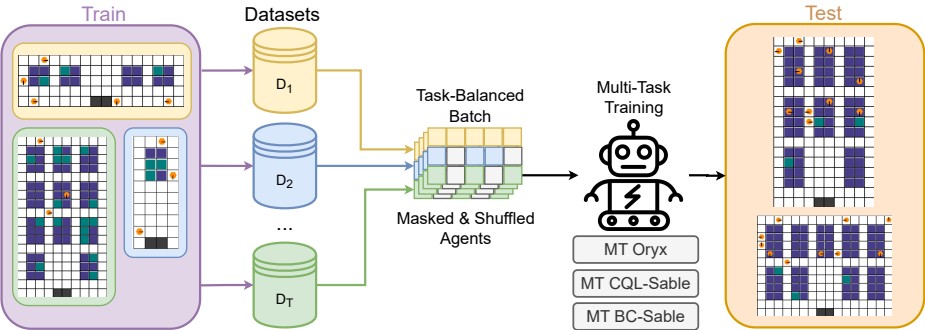

Figure 2: *Our offline multi-task multi-agent training and testing setup.* In this setup, there is a set of training tasks, each with a static dataset of pre-collected trajectories that together form a diverse multi-task dataset. This dataset is then used for training, without any additional online interactions with either the training tasks or the testing tasks. At evaluation time, the trained model is evaluated on each of the held-out test tasks, and the average test performance is calculated. In this illustration we used RWARE tasks in the train and test sets.

## 2 MULTI-TASK SEQUENCE MODELLING FOR OFFLINE MARL

### 2.1 PRELIMINARIES

**Problem formulation.** We formalise a cooperative MARL *task* as a Decentralised Partially Observable Markov Decision Process (Dec-POMDP) (Kaelbling et al., 1998), defined by the tuple $\mathcal{M}_\dagger = \langle \mathcal{N}, \mathcal{S}, \mathcal{A}, P, R, \{\Omega^i\}_{i \in \mathcal{N}}, \{E_i\}_{i \in \mathcal{N}}, \gamma \rangle$, where $\dagger$ denotes the particular task selected from an environment. For example, in a simulated robotic warehouse environment, a task corresponds to a specific warehouse layout and the number of robotic workers collecting and depositing requested shelf items. At each timestep $t$ within a task, the environment is in state $s_t \in \mathcal{S}$. Each agent $i \in \mathcal{N}$ selects an action $a_t^i \in \mathcal{A}^i$ based on its local action-observation history $\tau_t^i = (o_0^i, a_0^i, \ldots, o_t^i)$. The agents' actions form a joint action $\boldsymbol{a}_t \in \mathcal{A} = \prod_{i \in \mathcal{N}} \mathcal{A}^i$, which, when executed, yields a shared reward $r_t = R(s_t, \boldsymbol{a}_t)$, transitions the environment to $s_{t+1} \sim P(\cdot|s_t, \boldsymbol{a}_t)$, and provides each agent $i$ with a new observation $o_{t+1}^i \sim E_i(\cdot|s_{t+1}, \boldsymbol{a}_t)$. The agent then updates its history as $\tau_{t+1}^i = (\tau_t^i, a_t^i, o_{t+1}^i)$. The task-specific objective is to learn a joint policy $\pi(\boldsymbol{a}|\boldsymbol{\tau})$ that maximises the expected discounted return over a horizon of timesteps $H$: $J_\dagger(\boldsymbol{\pi}) = \mathbb{E}_\pi \left[ \sum_{t=0}^{H} \gamma^t r_t \right]$.

To create our train-test evaluation setup, we consider offline datasets $\mathcal{D}_{\text{train}} = \{\mathcal{D}_\dagger : \dagger \in \mathcal{T}_{\text{train}}\}$ collected from a set of training tasks $\mathcal{T}_{\text{train}}$. Our objective is to learn a single joint policy $\boldsymbol{\pi}_{\text{train}}$, using only the fixed multi-task training data (i.e. without any additional online interaction), to maximise the expected zero-shot performance on a set of *unseen* test tasks $\mathcal{T}_{\text{test}}$, given as

$$J(\boldsymbol{\pi}) = \mathbb{E}_{\dagger \sim \mathcal{T}_{\text{test}}}[J_\dagger(\boldsymbol{\pi})|\boldsymbol{\pi} = \boldsymbol{\pi}_{\text{train}}].$$

By optimising the above objective, we are minimising the generalisation gap between training and test tasks. A simplified visual representation of the problem setting is depicted in Figure 2.

**Multi-Agent Sequence Models.** Centralised control, where a single policy outputs the joint action, is theoretically optimal but scales poorly due to an exponential growth of the action space (de Kock et al., 2025). However, autoregressive factorisation is an efficient way to parametrise the joint policy, by expressing the joint distribution over $n$ agents as a product of conditional distributions:

$$\pi(\boldsymbol{a}|\boldsymbol{\tau}) = \prod_{k=1}^{n} \pi^{i_k}\left(a^{i_k} \mid \boldsymbol{\tau}, a^{i_1}, \ldots, a^{i_{k-1}}\right).$$

Here $i_k$ denotes an agent index from an ordered set $\{i_1, \ldots, i_n\} \in S_n$, where $S_n$ is the set of permutations of $\{1, ..., n\}$. This factorisation decomposes joint decision-making into a sequence of conditional actions, enabling scalable coordination, efficient parallel training and, in certain cases, providing desirable convergence properties (Zhong et al., 2024b). Sequence models provide a natural parameterisation of such policies, closely mirroring the autoregressive next token prediction

process in text and image generation, and have been demonstrated to work well on a large range of MARL settings (Wen et al., 2022; Mahjoub et al., 2025; Daniel et al., 2024; Formanek et al., 2025).

## 2.2 Multi-Task Sequence Models for Offline MARL

Building on existing multi-agent sequence models for offline MARL (Formanek et al., 2025), we propose a few simple yet essential modifications that enable training on multiple tasks with varying numbers of agents simultaneously, while allowing seamless zero-shot transfer. By design, our multi-task sequence models do not receive explicit task IDs or have task specific output heads, since this would limit their zero-shot transferability to new tasks. Instead, our models have to infer task information from observations, agent counts, and environment dynamics.

**Dynamic agent padding, shuffling and masking.** In order to dynamically handle variable numbers of agents across tasks, we zero-pad the inputs for absent agents and mask their contributions in the loss. Moreover, we randomise the ordering of both active and inactive agents at each training update, which encourages the model to share representations and transfer knowledge across agents.

**Multi-task training loss.** Given a set of training tasks $\mathcal{T}_{\text{train}} = \{\dagger_1, \ldots, \dagger_M\}$, with offline buffers $\{\mathcal{D}_\dagger\}_{\tau \in \mathcal{T}_{\text{train}}}$, we train a multi-task sequence model by minimizing the average per-task loss

$$\min_\theta \quad \frac{1}{M} \sum_{\dagger \in \mathcal{T}_{\text{train}}} \Big[ \mathcal{L}(\theta; \mathcal{D}_\dagger) \Big]. \tag{1}$$

The loss $\mathcal{L}$ changes depending on the algorithm used, which in our case includes autoregressive versions of behaviour cloning (BC) (Pomerleau, 1988; Bain & Sammut, 1995), Conservative Q-learning (CQL) (Kumar et al., 2020) and Implicit Constraint Q-learning (ICQ) (Yang et al., 2021b; Formanek et al., 2025).

**Task-balanced batching.** For each training update, we build a single unified mini-batch by evenly sampling across different tasks. Given a batch size $B$, we compute $q = \lfloor B / |\mathcal{T}_{\text{train}}| \rfloor$ and $r = B - q|\mathcal{T}_{\text{train}}|$. Each task $\dagger \in \mathcal{T}_{\text{train}}$, contributes $q$ samples; the remaining $r$ samples are assigned by round-robin across tasks up to the value $r$. This yields stochastic gradients that are unbiased over a uniform mixture of tasks (each task equally weighted), rather than a size-weighted mixture. The resulting task-balanced batching also mitigates "head-task" dominance seen with dataset-proportional sampling, a known issue in domain generalisation from long-tailed datasets (Cui et al., 2019).

**Value function learning via classification.** To mitigate gradient interference from varying reward scales across tasks, we replace scalar TD regression with a classification objective. Specifically, we use HL-Gauss (Imani & White, 2018; Farebrother et al., 2024), which projects each scalar TD target onto a discrete support by smoothing with a Gaussian distribution, and trains the value function with categorical cross-entropy over the resulting histogram. This choice, consistent with prior multi-task training architectures (Kumar et al., 2022a), improves stability and reduces loss-scale sensitivity compared to mean squared error.

## 2.3 Theoretical Underpinnings for Improved Generalisation

We posit that the superior zero-shot generalisation of our multi-task sequence models stems from their ability to perform *amortised Bayesian inference* (Gershman & Goodman, 2014) over a latent task space. We assume each task is governed by an unobserved latent variable $z \in \mathcal{Z}$ (the "task embedding"), which compactly parametrizes the transition dynamics and reward function (e.g., specifying map density or agent capabilities). Unlike single-task policies that overfit to a fixed $z_{train}$, a sequence model trained on a distribution of tasks implicitly learns an inference mapping $q_\phi(z \mid \tau_{1:t})$ from interaction history to task belief states (Xie et al., 2022). Crucially, in the multi-agent setting, accurate inference of $z$ requires aggregating partial information distributed across the team. Our sequence architecture facilitates this by jointly processing the observations and actions of all agents within the encoder and autoregressive decoder. This shared global context allows the model to pool evidence from the entire team's history to correctly infer task parameters (such as the total number of agents) that remain unobservable to fully independent policies. The policy then effectively acts by marginalizing over this inferred embedding:

$$\pi_\theta(\mathbf{a_t} \mid \tau_{1:t}) \approx \int_{\mathcal{Z}} \pi_\psi(\mathbf{a_t} \mid s_t, z) \, q_\phi(z \mid \tau_{1:t}) \, dz \tag{2}$$

By optimizing the negative log-likelihood across diverse tasks, the model is forced to identify these task-specific parameters directly from the context window. We hypothesise that increasing *model capacity* is critical here, as it reduces the *inference error* by allowing the network to approximate this complex posterior distribution more accurately.

Furthermore, we argue that the combinatorial nature of multi-agent systems (Mahajan et al., 2022) requires a specific form of robustness we term *marginal consistency*. Standard training often leads to brittle "Hero" dynamics where agents over-rely on specific team members. Our use of agent masking and shuffling fundamentally alters this by training the model on the power set of agent sub-coalitions. This enforces a constraint where the autoregressive decoder must yield a valid optimal policy for any subset of agents $\mathcal{C}$:

$$\mathcal{L}(\theta) \approx \mathbb{E}_{z \sim p(z)} \sum_{\mathcal{C}} \mathbb{E}_{\tau, \mathbf{a}_{\mathcal{C}} \sim \mathcal{D}_z} \left[ -\log \pi_\theta(\mathbf{a}_{\mathcal{C}} \mid \tau, \mathbf{a}_{\backslash \mathcal{C}}) A(\tau, \mathbf{a}_{\mathcal{C}}) \right] \tag{3}$$

This enables the model to act as a *flexible coalition coordinator*, capable of deriving robust cooperative strategies for team subsets of varying sizes. We expect this mechanism to significantly reduce the *coverage error* by densifying the effective training support, ensuring that the learned coordination primitives remain valid even when team compositions or sizes change in the test set.

Formally, we view the generalisation gap (regret) on a held-out task $z_{\text{test}}$ as the sum of these two distinct error terms. Following our derivation in Appendix A, the regret is bounded by:

$$\mathcal{R}(z_{\text{test}}) \leq \underbrace{C_1 \cdot \mathbb{E}_{z \sim q_\phi}[\|z_{\text{test}} - z\|]}_{\text{Inference}} + \underbrace{C_2 \cdot \min_{z_i \in \mathcal{D}} \|z_i - z_{\text{test}}\|}_{\text{Coverage}} \tag{4}$$

The *inference error* is bounded by the expected geometric distance (Wasserstein-1) between the inferred task belief and the ground truth, capturing the precision of the model's internal estimation. The *coverage error* measures the geometric distance between the test task and the nearest training task, representing the density of the training manifold. We provide the full derivation and detailed theoretical analysis of these bounds in Appendix A. Based on this decomposition, we expect that *task diversity* (densifying $\mathcal{D}$) will be the primary driver for reducing coverage error, while *model scale* (improving the approximation of $q_\phi$) will be the primary driver for reducing inference error.

## 3 EXPERIMENTS

### 3.1 EXPERIMENTAL DESIGN

**Tasks.** We considered three challenging MARL environments, `LBF`, `RWARE` (Papoudakis et al., 2021) and `Connector` (Bonnet et al., 2024). These are all widely used MARL benchmarks, with `RWARE` also proposed as a suitable multi-task benchmark in previous work (Schäfer, 2022) and `Connector` being of particular interest due to its agent scaling properties Formanek et al. (2025). For each environment, we selected several different level configurations to serve as distinct tasks. These tasks were then partitioned into train and test sets (see Appendix J), taking care to ensure that the test tasks were different in meaningful ways to the training tasks, as shown in Figure 3.

**Datasets.** For each task, we construct an offline dataset $\mathcal{D}_\dagger$ by recording a set of rollouts at fixed intervals from an online training run of SABLE (Mahjoub et al., 2025), a state-of-the-art MARL sequence model. This yields a mixed dataset with the same number of rollouts per task but not necessarily the same number of transitions, since episode lengths differ across tasks, hence the necessity for task-balanced batching. Observations and actions are standardised per environment. For sequence modeling, we sample fixed-length trajectory chunks (context length reported with other hyperparameters in Appendix K). Rewards are left unclipped during training and for comparability across tasks, we report normalised returns, where each task's episode return is normalised by the final episode return achieved by the online system on that task.

**Algorithms.** The main algorithm we consider is an adapted version of Oryx (Formanek et al., 2025), which we modify for multi-task training. As described in section 2, this includes (i) dynamic padding, masking and agent shuffling, (ii) task-balanced batching, and (iii) value learning using HL-Gauss (Farebrother et al., 2024). We refer to this version of Oryx as Multi-Task (MT) Oryx. In

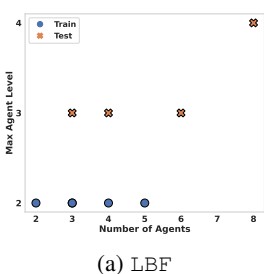 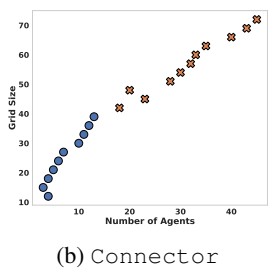 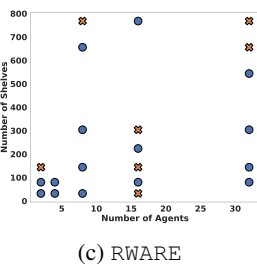

(a) `LBF`          (b) `Connector`          (c) `RWARE`

Figure 3: *Distributional shift between train and test tasks.* Each point represents a task with the number of agents in each task plotted against a specific task property: in `LBF`, the maximum agent level, in `Connector` the grid size, and in `RWARE` the number of shelves. While these dimensions are important to distinguish tasks, it should be noted there are additional parameters which change across tasks, not shown here (e.g. the layout of shelves in `RWARE` tasks).

addition, we develop two new strong baselines. The first is MT BC-Sable, which is an offline variant of Sable that uses simple behaviour cloning to train an autoregressive policy, along with dynamic padding and masking of agents, and task-balanced batching. The second is MT CQL-Sable, another offline variant of Sable that uses an autoregressive version of the CQL loss (Kumar et al., 2020), along with all three MT enhancements as in MT Oryx. The Sable network backbone is consistent across all three algorithms. Therefore, the only significant difference between MT Oryx and the other two baselines is the loss function $\mathcal{L}$ used. We chose CQL because of its proven generalisation and scaling capabilities in the single-agent setting (Kumar et al., 2022a; Chebotar et al., 2023), and BC for its competitive generalisation performance as demonstrated in prior work (Mediratta et al., 2024). Hyperparameter details for all three algorithms are listed in Appendix K.

**Evaluation protocol.** In our experiments, we are interested in the expected zero-shot performance of the trained model on the held-out test tasks. To measure this, we compute the absolute episode return (Gorsane et al., 2022), by running the best checkpoint achieved during training for 320 independent evaluation episodes and averaging the episode returns for each task in the test set. To compare across tasks and environments with potentially different reward scales, we normalise the absolute episode return by dividing it by the maximum expected episode return achieved on the respective task by the online Sable algorithm. Each run configuration was repeated across three random seeds, with the mean and standard deviation being reported in each case.

### 3.2 MULTI-TASK TRAINING IMPROVES GENERALISATION

**Experiment.** We vary the number of tasks in the training set, while keeping the test set fixed. We then train our multi-task sequence models on different subsets of the training datasets and measure the performance on the test tasks. For `LBF`, we consider a total of 5 training tasks, for `Connector` 10 and for `RWARE` 15, incrementing training by a single task from 1 to the maximum for each environment. We plot the performance across training task counts when evaluated on the same training tasks as well as the held-out test tasks in Figure 4.

**Discussion.** We observe that performance on the training tasks remains relatively high across all environments, even as the number of tasks increases. This indicates that the model can successfully learn across multiple tasks simultaneously. However, in `RWARE` we note a progressive decline in training performance as the number of training tasks grows. We attribute this to the higher complexity of `RWARE` tasks and the need to scale model capacity with task diversity to maintain performance. Interestingly, even as train task performance degrades, test task performance improves nearly monotonically as the number of training tasks increases, highlighting the importance of diverse multi-task data for generalisation. On `LBF`, we observe that MT CQL-Sable's performance decreases. We hypothesise that this is due to the high proportion of expert trajectories in the `LBF` dataset, as the data collection policy quickly converges to the optimal behaviour. Prior work has shown that CQL is particularly sensitive to overly narrow or high-quality datasets, and benefits from mixed quality datasets (Schweighofer et al., 2022). To further examine this, we include an ablation on trajectories' quality in Appendix D.

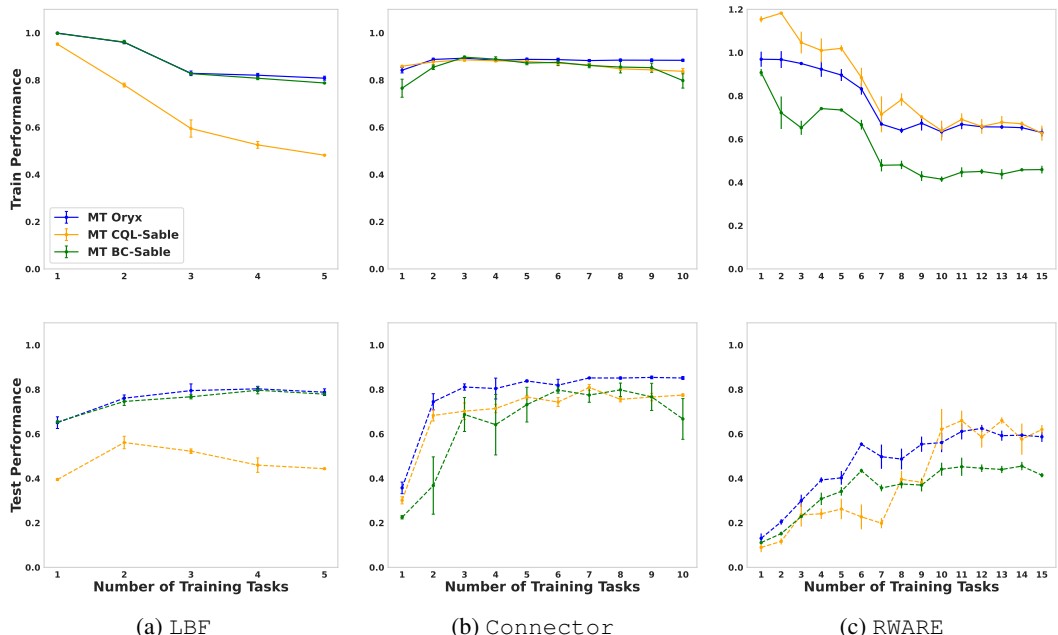

Figure 4: *The effect of increasing task diversity on performance.* **Top:** training tasks. **Bottom:** held-out test tasks. When we train our sequence models using only a single task, we observe strong performance on that single training task (see first point on each plot in the top row). However, the performance on the held-out test tasks is much lower, i.e. the generalisation gap is large. **As we increase the number of tasks in the training set, we observe a steady increase in the test task performance across all three environments.**

Across all algorithms and environments, performance tends to plateau after a certain number of training tasks. We attribute this saturation to the limits of the current model capacity, pointing to the necessity of scaling up the model size to obtain maximum performance on highly diverse multi-task datasets (see subsection 3.3). To summarise the overall effect of multi-task training with a fixed model size, we measure and report the maximum performance gain on test tasks in Figure 1. Averaged across all three algorithms, test performance improves by 5.4x on RWARE, 1.3x on LBF, and 2.9x on Connector. These results validate the effectiveness of multi-task training as a means of unlocking substantial performance gains on unseen test tasks.

### 3.3 CAN WE FURTHER IMPROVE GENERALISATION BY INCREASING THE SIZE OF THE DATASETS AND MODELS?

A natural question that arises is what is the optimal dataset size and model size for generalisation. Can we improve the generalisation capabilities by simply increasing the size of the dataset for a given set of training tasks? Similarly, can we improve generalisation by increasing the size of the model? To test this we design two experiments.

**Experiment (a).** To determine whether increasing the size of the datasets (in terms of number of transitions rather than number of tasks helps performance) we conducted a sweep over dataset sizes for several multi-task datasets on RWARE. The results of the sweep are presented in Figure 5a. Similar to the results by Mediratta et al. (2024), we find that there is little evidence that scaling up the number of transitions helps generalisation nearly as much as adding more tasks.

**Experiment (b).** To study the effect of model size, we train various models with different numbers of parameters, ranging from 116k to 13M, using the RWARE dataset. For simplicity, we mainly vary the embedding dimension of the model's encoder-decoder network from 64 (116k parameters) to 768 (13M parameters). We report the average episode return, normalised by the online performance, on both the training and test tasks in Figure 5b. We show in Appendix E similar results for LBF and Connector.

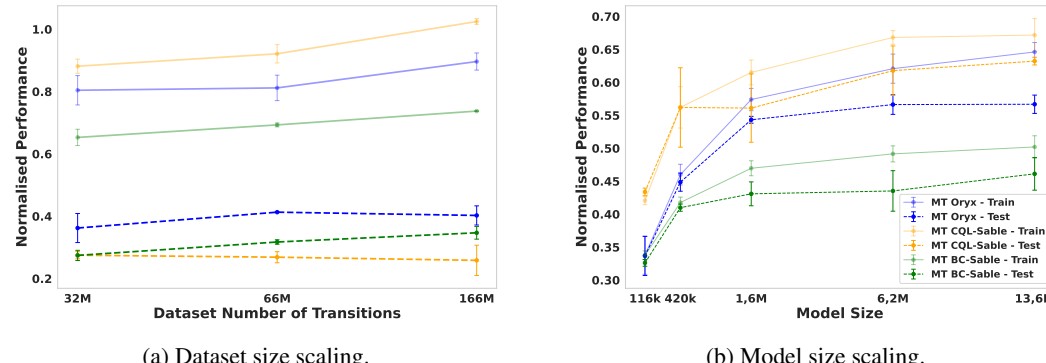

(a) Dataset size scaling.

(b) Model size scaling.

Figure 5: *The impact of scaling up dataset (**left**) and model size (**right**).* When we fix the number of `RWARE` tasks in the dataset to 5 but grow the number of transitions in the dataset, we observe an increase in train performance, while the test performance plateaus. On the other hand, when we train each of our MT sequence models on the full 15 task `RWARE` dataset, **we observe a clear scaling trend with respect to the model size in terms of both train and test performance.**

**Discussion.** The results in Figure 5a indicate that simply increasing the number of transitions in the training dataset improves train task performance but does not lead to better generalisation on held-out test tasks, highlighting the importance of task diversity in multi-task datasets, since from Figure 4c we can conclude that adding additional tasks has a greater benefit. In contrast, scaling model capacity (Figure 5b)—from an embedding dimension of 64 (116k parameters) to 512 (6.2M parameters)—consistently improved both training and test performance. This finding is particularly encouraging: it suggests that large, diverse multi-task datasets may be the missing ingredient needed to make ever-larger and more general offline MARL models viable. Notably, this result contrasts with the single-task setting reported by (Formanek et al., 2025), where the optimal embedding dimension was just 64, underscoring the unique potential of multi-task data for enabling scale.

## 3.4 ABLATION STUDIES

**HL-Gauss.** To test the effect of using HL-Gauss (Farebrother et al., 2024) for multi-task learning, we conduct an ablation on the full set of `RWARE` training tasks where we run MT Oryx and MT CQL-Sable with and without HL-Gauss for value function learning (e.g. standard TD mean-squared-error). We compare the algorithms on multi-task `RWARE` since the task-to-task variance in episode returns is significant and therefore more challenging to accurately learn a multi-task value function. As shown in Figure 6a, using HL-Gauss leads to slightly better performance ($\approx 8\%$ improvement) on test tasks for MT Oryx, while the effect on MT CQL-Sable is marginal.

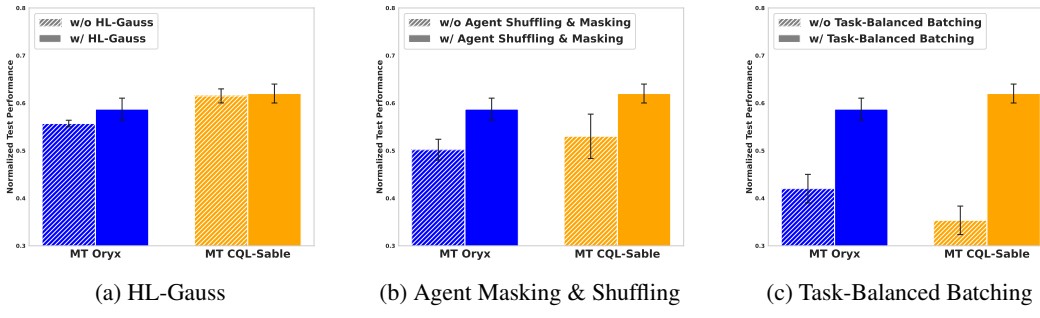

(a) HL-Gauss

(b) Agent Masking & Shuffling

(c) Task-Balanced Batching

Figure 6: *Ablation studies.* **Left:** Using HL-Gauss improves test performance for MT Oryx by $\approx 8\%$, while the effect on MT CQL-Sable is marginal. **Middle:** Disabling agent masking and shuffling reduces test performance by $\approx 16\%$ on average for both algorithms. **Right:** Removing task-balanced batching has the highest impact with $\approx 37\%$ drop in test performance on average for both MT Oryx and MT CQL-Sable.

**Agent shuffling and masking.** To test the impact of not masking and shuffling agents we conduct a similar ablation to above on RWARE. We observe decrease in performance of ≈ 16% on average for both algorithms on the test tasks, when we do not mask and shuffle agents (see Figure 6b).

**Task-balanced batching.** Finally, we conducted an ablation on how we sample data from the multi-task dataset. In the first case we use our proposed task-balanced batching method, which includes a fair mix of samples from each task in every batch. In the alternative approach we choose a random task at each update step and sample a full batch from the chose single task. The results in Figure 6c shows a 37% decrease in test performance on average for both MT Oryx and MT CQL-Sable without task-balanced batching.

### 3.5 HOW DOES MT ORYX COMPARE TO PRIOR WORK?

In orderd to establish how our MT Sequence models compare to prior multi-task MARL methods we evaluated our best model (see Appendix C), MT Oryx, on the SMAC datasets and tasks from Zhang et al. (2023a) and compared it to their method called ODIS. The results of which are shared in Table 1 and show that our method compares well in terms of zero-shot transfer to unseen SMAC maps. Further to this, we tested if our task scaling result held across their SMAC datasets. We scaled from a single task to their full set of three training tasks in the marine-hard task set using the expert datasets. As we can see from Figure 7, similar scaling trends hold. In order to handle the varying action and observation spaces across tasks we used the same decomposition strategy used by Zhang et al. (2023a).

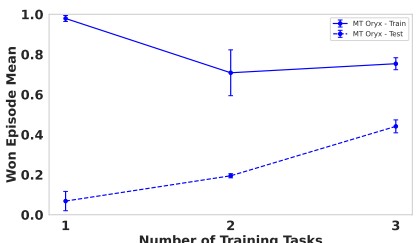

Figure 7: Increasing the number of training tasks on SMAC tends to increase test performance.

Table 1: MT Oryx vs. ODIS on the SMAC test suite, Marine-Hard (Zhang et al., 2023a). Bold indicates the highest mean and * indicates no statistical difference ($p \geq 0.05$) using a two-sided t-test (Papoudakis et al., 2021).

| Split | Tasks | Expert Data | | Medium Data | | Medium-Replay | |
|---|---|---|---|---|---|---|---|
| | | ODIS | MT Oryx | ODIS | MT Oryx | ODIS | MT Oryx |
| Train | 3m | **98.4 ± 2.7** | 94.8 ± 3.6* | **85.9 ± 10.5** | 52.1 ± 9.5 | **83.6 ± 14.0** | 47.9 ± 16.0 |
| | 5m6m | 53.9 ± 5.1* | **55.2 ± 6.5** | 22.7 ± 7.1* | **22.9 ± 15.7** | 16.6 ± 4.7* | **18.8 ± 3.1** |
| | 9m10m | 80.4 ± 8.7* | **89.6 ± 7.2** | **78.1 ± 3.8** | 29.2 ± 10.0 | **34.4 ± 8.0** | 14.6 ± 9.5 |
| Test | 4m | **95.3 ± 3.5** | 64.6 ± 27.3* | **61.7 ± 17.7** | 43.8 ± 11.3* | 55.6 ± 14.5* | **57.3 ± 4.8** |
| | 5m | **89.1 ± 10.0** | 86.5 ± 11.8* | 85.9 ± 11.8* | **99.0 ± 1.8** | **96.1 ± 4.1** | 95.8 ± 1.8* |
| | 10m | 93.8 ± 2.2 | **100.0 ± 0.0** | 61.3 ± 11.3 | **83.3 ± 6.5** | 84.4 ± 15.1* | **90.6 ± 8.3** |
| | 12m | 58.6 ± 11.8* | **77.1 ± 19.1** | 35.9 ± 8.1 | **72.9 ± 3.6** | **84.4 ± 6.6** | 66.7 ± 3.6 |
| | 7m8m | **25.0 ± 15.1** | 3.1 ± 3.1 | **28.1 ± 22.0** | 4.2 ± 3.6* | **9.4 ± 2.2** | 7.3 ± 3.6* |
| | 8m9m | 19.6 ± 6.0* | **20.8 ± 13.0** | 4.7 ± 2.7* | **8.3 ± 6.5** | **11.7 ± 8.7** | 11.5 ± 7.9* |
| | 10m11m | 42.4 ± 7.2* | **64.6 ± 21.3** | **29.7 ± 15.4** | 15.6 ± 8.3* | **35.9 ± 5.2** | 14.6 ± 4.8 |
| | 10m12m | 1.6 ± 1.6 | **6.2 ± 0.0** | **1.6 ± 1.6** | 0.0 ± 0.0* | **2.3 ± 1.4** | 0.0 ± 0.0 |
| | 13m15m | **2.3 ± 2.6** | 2.1 ± 1.8* | **1.6 ± 1.6** | 0.0 ± 0.0* | **2.4 ± 1.4** | 0.0 ± 0.0 |

## 4 RELATED WORK

**Offline MARL.** Most prior work in offline MARL uses single-task training and evaluation, while focusing on finding solutions to key challenges particular to offline multi-agent learning. Seminal early papers include Jiang & Lu (2021) and Yang et al. (2021a), who introduced multi-agent methods for constrained Q-value estimation. Since then, numerous additional works have aimed to tackle challenges such as extrapolation error (Shao et al., 2023; Eldeeb et al., 2024), coordination (Barde et al., 2024; Tilbury et al., 2024; Zhou et al., 2025), offline training stability (Pan et al., 2022; Wang et al., 2023; Matsunaga et al., 2023; Wu et al., 2023a; Bui et al., 2025; Liu et al., 2024b; Li et al., 2025), opponent modeling (Jing et al., 2024), offline-to-online transfer (Zhong et al., 2024a; Formanek et al., 2023) and theoretical understanding (Cui & Du, 2022b;a; Zhong et al., 2022; Zhang et al., 2023b; Xiong et al., 2023; Wu et al., 2023a).

**Sequence Models for RL.** Formulating RL as a sequence modelling problem has gained significant attention. Chen et al. (2021) introduced the Decision Transformer (DT), later extended in various ways (Zheng et al., 2022; Yamagata et al., 2023; Wu et al., 2023b). Lee et al. (2022) trained a multi-task DT that learned across tasks and could be quickly fine-tuned. Meng et al. (2023) introduced MADT, an extension of the DT to the multi-agent setting. The Multi-Agent Transformer (MAT) (Wen et al., 2022) addressed the online setting with auto-regressive action selection, and Mahjoub et al. (2025) improved on MAT with Sable, which replaces the Transformer with a Retentive Network (Sun et al., 2023) and adds temporal memory, achieving state-of-the-art results. Building on this line, Formanek et al. (2025) proposed Oryx, an offline MARL sequence model derived from an autoregressive version of Implicit Constraint Q-Learning (ICQ) (Yang et al., 2021b) and offline-specific modifications to Sable, also achieving state-of-the-art performance.

**Multi-Task RL.** Multi-task training has most prominently been investigated in single-agent continuous-control and robotics problems with a focus on representation and transfer learning (Xu et al., 2020; Kalashnikov et al., 2021; Kumar et al., 2022b; Cheng et al., 2022). Although shown to be useful in most cases, Yu et al. (2021) find that naively adding more multi-task data to an offline RL training dataset can sometimes lead to a decrease in performance on downstream tasks, particularly when the distributional shift between tasks is large. In terms of generalisation, Kumar et al. (2022a) and He et al. (2023) highlight the potential for high-capacity models trained on large and diverse multi-task datasets to produce agents that can generalise more broadly when fine-tuned on previously unseen tasks. Most closely related to our work is that of Mediratta et al. (2024), who evaluate the zero-shot generalisation capabilities of several offline single-agent RL methods by training them on a set of training tasks and testing them on a set of holdout tasks. They find that current offline RL methods do not generalise well and are typically outperformed by simple behaviour cloning.

**Multi-Task MARL.** Multi-task MARL faces both architectural and evaluation challenges when agents must generalise beyond single-task training, motivating formal definitions and benchmarks for task generalisation(Schäfer, 2022). Rosen et al. (2024) give a formal, goal-oriented theory that proves how a learned world value function can enable provably optimal zero-shot task generalisation in goal-based multi-agent settings. MaskMA (Liu et al., 2024a) introduces a mask-based framework that adapts to varying agent- and action-spaces and shows strong zero-shot transfer on unseen SMAC (Samvelyan et al., 2019) maps. Unlike our approach, their work builds on MADT (Meng et al., 2023), while we focus on sequence model architectures related to Oryx (Formanek et al., 2025), which have been shown to outperform MADT. The offline coordination-skill discovery method ODIS (Zhang et al., 2023a) extracts task-invariant coordination primitives from multi-task trajectories and shows that this can be used to deploy coordination policies to unseen SMAC tasks without additional online interaction. Related work, HiSSD (Liu et al., 2025) proposes a hierarchical separation between common cooperative (temporal) skills and task-specific controllers. None of the above studies investigates the effect of task diversity on test performance, instead keeping the number of training tasks fixed.

## 5 Conclusion

In this work, we studied generalisation in offline MARL and showed that task diversity is a key driver of improved test performance. We introduced a simple yet effective recipe for building multi-task sequence models, which consistently narrows the train–test gap and achieves significant performance gains on unseen test tasks. Our findings suggest that future progress in offline MARL should prioritise (i) constructing large and diverse, multi-task datasets, and (ii) carefully tuning their models' capacity for the given data budget to maximise zero-shot generalisation. We release code, datasets, task splits, and training scripts to encourage reproducibility and to establish stronger benchmarks for evaluating generalisation in offline MARL.

**Limitations and future work.** Our work is limited to centralised sequence model architectures, and although these represent a powerful and performant model class, promising future work could include extending our analysis to decentralised and CTDE algorithms. Additional areas of interest include studying the limits of transfer across environments (not only tasks), and investigating accelerating fine-tuning in safety-critical and data-scarce real-world domains.

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

LIST OF APPENDICES

## A    EXPANDED THEORETICAL ANALYSIS OF MULTI-TASK GENERALISATION

We formally justify the generalisation capabilities of our multi-task sequence models by viewing the training process as amortised Bayesian inference (Gershman & Goodman, 2014) over a latent task space, regularised by marginal consistency constraints imposed by the autoregressive structure and agent masking. We corroborate this analysis with specific empirical evidence from our experiments.

### A.1    LATENT TASK INFERENCE VIA SEQUENCE MODELLING

Consider a family of multi-agent tasks defined by a latent variable $z \in \mathcal{Z}$, where each $z$ specifies the transition dynamics $P(\cdot|s, a, z)$ and reward function $R(s, a, z)$ for a team of agents. In the offline setting, the dataset $\mathcal{D}$ consists of trajectories generated under various $z \sim p(z)$. While our model is trained via a standard supervised objective to minimize the negative log-likelihood (NLL) of actions, we posit that this is equivalent to maximizing the Evidence Lower Bound (ELBO). Since the optimal action depends on the unobserved task variable $z$, the model must implicitly infer $z$ to minimize NLL. This optimization forces the model to learn an internal inference network $q_\phi(z|\tau_{1:t})$ mapping history to task beliefs, alongside a policy $\pi_\psi(a|s, z)$.

Crucially, unlike fully independent policies that must infer task context solely from local observations, our sequence model explicitly models the relational structure between all agents' observations in the encoder and their actions in the decoder. This joint processing allows the inference network $q_\phi$ to aggregate evidence across the entire team, significantly enhancing task inference capacity, particularly in scenarios with varying agent counts where global context is emergent rather than local. We characterise this process as amortised inference because the heavy computational cost of calculating the complex posterior distribution is "amortised" (paid upfront) during the extensive training phase. Consequently, at test time, the model does not need to run expensive optimization algorithms; it simply performs fast, implicit inference via a single forward pass of the network. Following the framework of Xie et al. (2022), the sequence model approximates the posterior predictive distribution:

$$\pi_\theta(a_t|\tau_{1:t}) \approx \int_{\mathcal{Z}} \pi_\psi(a_t|s_t, z)q_\phi(z|\tau_{1:t})dz. \tag{5}$$

In single-task (ST) training, the prior $p(z)$ collapses to a Dirac delta $\delta_{z_{train}}$ causing $q_\phi$ to degenerate and ignore the history $\tau_{1:t}$. Conversely, multi-task (MT) training forces $q_\phi$ to extract task-identifying features from the context window.

*Note:* In Equation 5, the learned policy $\pi_\theta$ conditions on the history $\tau$ (resolving partial observability), while the implicit oracle $\pi_\psi$ is defined on the underlying state $s$ and true task $z$. The inference network $q_\phi$ bridges this gap by mapping history to a belief over $z$.

This theoretical view is strongly supported by our scaling results. We observe that increasing task diversity (the support of $p(z)$) leads to continuous improvements in zero-shot performance (Figure 4), whereas simply scaling dataset size for a fixed number of tasks yields diminishing returns (Figure 5a). This confirms that reducing the coverage error of the latent manifold is the primary driver of generalisation. Furthermore, the finding that performance scales with network capacity (Figure 5b) suggests that larger models are necessary to accurately approximate the complex inference posterior $q_\phi$. Qualitatively, the distinct strategies observed in RWARE where the model infers collision-avoidance behaviours in congested maps versus exploration in sparse maps (Figure 19, Figure 20) demonstrate successful context-driven inference of $z$.

### A.2    COMBINATORIAL GENERALISATION VIA MARGINAL CONSISTENCY

A unique challenge in MARL is the combinatorial complexity of the joint action space, often requiring what Mahajan et al. (2022) term Combinatorial Generalisation. Our architecture decomposes the joint policy autoregressively as $\pi_\theta(a|s) = \prod_{i=1}^{n} \pi_\theta(a^{\sigma(i)}|s, a^{\sigma(<i)})$, where $\sigma$ is a random permutation. In standard ST training, the model overfits to specific correlations between fixed agents, often collapsing into a "Hero" dynamic where a small subset of agents dominates the policy.

However, our use of agent masking during MT training fundamentally alters this dynamic. By randomly masking subsets of agents, we enforce marginal consistency. In this context, "marginal" refers to the policy distribution of a subset of agents (integrating out the others), and "consistency"

ensures that the model's prediction for this subset remains valid and optimal even when isolated from the full team. Mathematically, the objective approximates minimizing the advantage-weighted KL divergence between the model's marginals and the data distribution for all sub-coalitions $\mathcal{C} \subseteq \{1, ..., n\}$:

$$\mathcal{L}(\theta) \approx \mathbb{E}_{z \sim p(z)} \sum_{\mathcal{C}} \mathbb{E}_{\tau \sim \mathcal{M}_z} [-\log \pi_\theta(a_\mathcal{C}|s, a_{\setminus \mathcal{C}})]. \tag{6}$$

This forces the autoregressive decoder to function as a coordination inference engine, learning permutation-invariant coordination primitives that are robust to variations in team composition.

Our ablation studies validate this mechanism, showing that removing agent masking and shuffling results in a $\approx 16\%$ drop in test performance (Figure 6b).

### A.3 FORMAL ERROR DECOMPOSITION

To analyse the generalisation gap, we define the regret on a test task $z_{\text{test}}$ as $\mathcal{R}(z_{\text{test}}) = V^{\pi^*}(z_{\text{test}}) - V^{\pi_\theta}(z_{\text{test}})$, where $V^{\pi^*}$ is the optimal value and $V^{\pi_\theta}$ is the value of our learned policy evaluated in the true task environment. We introduce an auxiliary "oracle" policy $\pi_\psi(\cdot|s, z_{\text{test}})$ which shares the learned control weights but receives the true task ID $z_{\text{test}}$ as input. Unlike the oracle, our learned policy $\pi_\theta$ operates on an inferred task embedding $\hat{z}$. Using the triangle inequality, we can decompose the regret into two terms:

$$\mathcal{R}(z_{\text{test}}) \leq \underbrace{|V^{\pi^*}(z_{\text{test}}) - V^{\pi_\psi}(z_{\text{test}})|}_{\epsilon_{\text{coverage}}} + \underbrace{|V^{\pi_\psi}(z_{\text{test}}) - V^{\pi_\theta}(z_{\text{test}})|}_{\epsilon_{\text{inference}}}. \tag{7}$$

*Remark:* In the second term, although both value functions are evaluated on the true task $z_{\text{test}}$, the divergence arises because $\pi_\psi$ conditions on $z_{\text{test}}$ while $\pi_\theta$ conditions on the inferred $\hat{z}$.

*Proof.* By adding and subtracting the oracle value term $V^{\pi_\psi}(z_{\text{test}})$ inside the regret definition and applying the triangle inequality:

$$\begin{aligned} \mathcal{R}(z_{\text{test}}) &= V^{\pi^*}(z_{\text{test}}) - V^{\pi_\theta}(z_{\text{test}}) \\ &= V^{\pi^*}(z_{\text{test}}) - V^{\pi_\psi}(z_{\text{test}}) + V^{\pi_\psi}(z_{\text{test}}) - V^{\pi_\theta}(z_{\text{test}}) \\ &\leq |V^{\pi^*}(z_{\text{test}}) - V^{\pi_\psi}(z_{\text{test}})| + |V^{\pi_\psi}(z_{\text{test}}) - V^{\pi_\theta}(z_{\text{test}})|. \end{aligned}$$

Here, $V^{\pi_\theta}(z_{\text{test}})$ denotes the value of the policy $\pi_\theta$ (which acts based on the inferred belief $\hat{z} \sim q_\phi(\cdot|\tau)$) when interacting with the true environment $z_{\text{test}}$. Consequently, the second term precisely measures the performance gap caused by acting upon the inferred representation $\hat{z}$ rather than the ground truth $z_{\text{test}}$. $\square$

**Coverage Error.** The coverage error $\epsilon_{\text{coverage}} = |V^{\pi^*}(z_{\text{test}}) - V^{\pi_\psi}(z_{\text{test}})|$ represents the approximation error of the shared policy manifold due to the finite support of the training distribution $\mathcal{D}$. Even if the task identity were known perfectly, this error persists if the training tasks do not sufficiently cover the task space. This formalises why task diversity (densifying the support of $\mathcal{D}$) is the primary driver for reducing this error, as confirmed by our results in Figure 4.

**Theorem 1** (Coverage Bound)**.** *Assuming the optimal value function is $L$-Lipschitz continuous with respect to the task metric $d(\cdot, \cdot)$ and the model fits the training tasks well ($\epsilon_{train} \approx 0$), the coverage error is bounded by the distance to the nearest training task:*

$$\epsilon_{coverage} \leq 2L \cdot \min_{z_i \in \mathcal{D}} d(z_i, z_{test}). \tag{8}$$

*Proof.* Let $z_{NN} = \arg\min_{z_i \in \mathcal{D}} d(z_i, z_{\text{test}})$. We decompose the error using the triangle inequality:

$$\epsilon_{\text{coverage}} \leq |V^{\pi^*}(z_{\text{test}}) - V^{\pi^*}(z_{NN})| + |V^{\pi^*}(z_{NN}) - V^{\pi_\psi}(z_{NN})| + |V^{\pi_\psi}(z_{NN}) - V^{\pi_\psi}(z_{\text{test}})|.$$

The first term is bounded by $L \cdot d(z_{\text{test}}, z_{NN})$ due to Lipschitz continuity. The second term vanishes under the assumption of successful training on the support set. The third term is similarly bounded by $L \cdot d(z_{\text{test}}, z_{NN})$ assuming the learned policy inherits the Lipschitz property. Summing these yields $2L \cdot d(z_{\text{test}}, z_{NN})$. $\qquad\square$

**Inference Error.** The inference error $\epsilon_{\text{inference}} = |V^{\pi_\psi}(z_{\text{test}}) - V^{\pi_\theta}(z_{\text{test}})|$ measures the identification gap caused by using the inferred posterior $q_\phi$ instead of the true ID. It captures the penalty for acting on an incorrect task belief $\hat{z} \sim q_\phi(\cdot|\tau)$. As noted by Ghosh et al. (2021), generalisation in RL often fails due to this "epistemic POMDP" problem where the posterior belief is misaligned. Minimizing this term requires sufficient model capacity to approximate the complex inverse mapping from trajectories to task parameters, explaining the scaling behaviour observed in Figure 5b.

**Theorem 2** (Inference Bound). *Let the oracle policy $\pi_\psi(\cdot|s, z)$ be $L_\pi$-Lipschitz continuous in $z$ with respect to the Total Variation (TV) distance. The inference error is bounded by:*

$$\epsilon_{\text{inference}} \leq C \cdot \mathbb{E}_{z \sim q_\phi}[\|z_{\text{test}} - z\|]. \tag{9}$$

*Proof.* We first apply the Value Difference Lemma to bound the gap between the oracle policy $\pi_\psi(\cdot|z_{\text{test}})$ and the inferred policy $\pi_\theta(\cdot) = \mathbb{E}_{z \sim q_\phi}[\pi_\psi(\cdot|z)]$ by the expected divergence in their action distributions:

$$|V^{\pi_\psi}(z_{\text{test}}) - V^{\pi_\theta}(z_{\text{test}})| \leq \frac{V_{max}}{1-\gamma} \mathbb{E}_{s \sim d^{\pi_\psi}}[D_{TV}(\pi_\psi(\cdot|s, z_{\text{test}})\|\pi_\theta(\cdot|s))]$$

We apply Jensen's inequality to the convex Total Variation distance function:

$$D_{TV}\left(\pi_\psi(\cdot|s, z_{\text{test}})\Big\|\mathbb{E}_{z \sim q_\phi}[\pi_\psi(\cdot|s, z)]\right) \leq \mathbb{E}_{z \sim q_\phi}[D_{TV}(\pi_\psi(\cdot|s, z_{\text{test}})\|\pi_\psi(\cdot|s, z))].$$

By the Lipschitz assumption on the policy with respect to the task parameter $z$:

$$\mathbb{E}_{z \sim q_\phi}[D_{TV}(\pi_\psi(\cdot|s, z_{\text{test}})\|\pi_\psi(\cdot|s, z))] \leq L_\pi \mathbb{E}_{z \sim q_\phi}[\|z_{\text{test}} - z\|].$$

The term $\mathbb{E}_{z \sim q_\phi}[\|z_{\text{test}} - z\|]$ is the 1-Wasserstein distance between the Dirac $\delta_{z_{\text{test}}}$ and the belief $q_\phi$. This confirms that minimizing the geometric distance in the latent space minimizes inference error. $\qquad\square$

*Remark:* While our practical implementation operates on interaction histories $\tau$ to handle partial observability, standard theoretical bounds are defined on the underlying Markovian state space $s$. This interchange is valid under the assumption that the sequence model acts as a belief state encoder. In the limit of sufficient capacity, the history $\tau$ serves as a *sufficient statistic* for the state $s$ and the task belief $q(z)$. Therefore, bounding the error over the state distribution $s \sim d^\pi$ implicitly bounds the performance of the history-based policy.

This decomposition aligns with recent theoretical frameworks in generalization and representation learning (Ghosh et al., 2021; Cheng et al., 2022), which identify epistemic uncertainty (Ghosh et al., 2021) and shared representation error (Cheng et al., 2022) as the two dominant sources of regret.

# B ENVIRONMENT DETAILS

## B.1 LBF

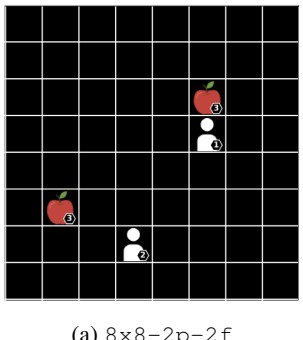 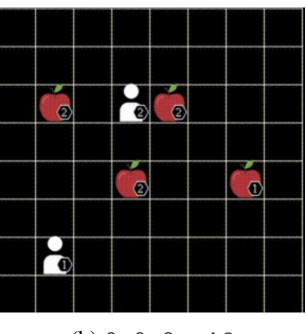

(a) `8x8-2p-2f`                    (b) `8x8-2p-4f`

Figure 8: LBF

In the Level-Based Foraging (LBF) environment, which is a JAX-based implementation from the Jumanji suite (Bonnet et al., 2024) of the original framework by Papoudakis et al. (2021), agents with assigned levels navigate a grid world to collect food items that can only be consumed if the sum of adjacent agent levels exceeds the food's level. These tasks are defined by the naming convention `<x size>x<y size>-<n agents>p-<food>f`, specifying the grid dimensions, agent and food counts. Agents observe a limited $5 \times 5$ square grid centered on their location which reveals the positions and levels of nearby items. Operating via a discrete action space of six options that includes no-operation, loading food, and movement in the four cardinal directions, agents receive rewards calculated as the sum of collected food levels divided by the level of the contributing agents.

## B.2 CONNECTOR

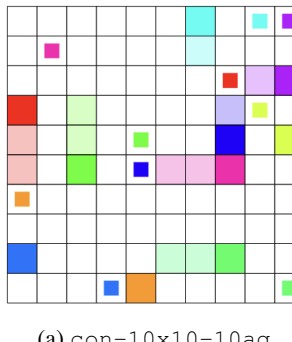 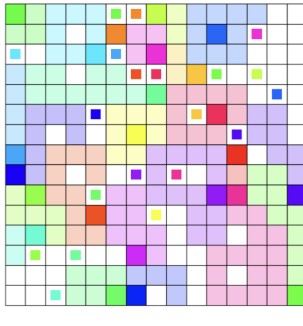

(a) `con-10x10-10ag`                    (b) `con-15x15-23ag`

Figure 9: Connector

In the Connector (Bonnet et al., 2024) environment, multiple agents are randomly initialized within a grid world to connect assigned start and end points in the minimum number of steps, a task complicated by the fact that movement creates permanent, impassable trails which necessitate cooperation to avoid blocking teammates. These tasks follow the naming convention `con-<x-size>x<y-size>-<num_agents>a` to specify grid dimensions and agent count. Agents operate within this system by observing an $n \times n$ local view centered on their location that reveals trails and all target locations, while also accessing the global $(x, y)$ coordinates of their current position and specific destination. Acting through a discrete space of five options including up, down, left, right, and stop, agents are guided by a reward function that yields $+1$ at the moment of connection and a penalty of $-0.03$ for every other step until completion.

## B.3   RWARE

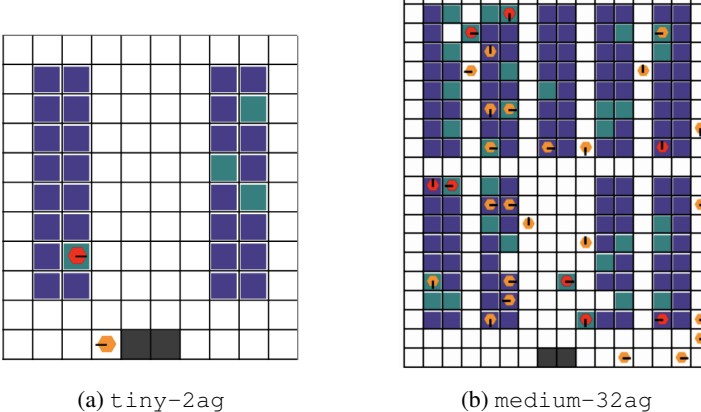

(a) `tiny-2ag`                    (b) `medium-32ag`

Figure 10: RWARE

The Robot Warehouse (RWARE) environment simulates a logistics scenario where a team of autonomous robots must fetch requested goods from shelves and deliver them to workstations to maximize throughput. We utilize the JAX-based implementation from the Jumanji suite (Bonnet et al., 2024) based on the original work by Papoudakis et al. (2021), which notably terminates episodes immediately upon agent collision rather than attempting to resolve the conflict. Tasks follow the convention `<size>-<num agents>ag`, where the size determines the shelf layout. Agents operate under partial observability within a $3 \times 3$ view centered on their position that reveals self and peer states alongside shelf status, using a discrete action space of five commands for navigation and loading to achieve a sparse reward of $+1$ granted solely for successful deliveries.

# C  MULTI-TASK OFFLINE MARL CAN GENERALISE BETTER THAN BEHAVIOUR CLONING

The findings from Mediratta et al. (2024) paint a bleak outlook for the generalisation capabilities of Offline RL algorithms compared to simple behaviour cloning. To establish if we observe a similar trend, we aggregate the normalised episode returns across all test tasks from `LBF`, `RWARE` and `Connector`, when trained using the full training set, to compare our three algorithms. In Table 2, we show the mean and standard error for each algorithm.

We want to know which offline training objective performed the best in terms of generalisation to the test tasks. We considered three objectives: behaviour cloning, conservative Q-learning, and the autoregressive ICQ loss from Formanek et al. (2025). We find that on LBF and Connector Oryx (ICQ loss) performs the best, followed by BC and then only CQL. On RWARE, on the other hand, CQL does the best, followed by ICQ and then BC. We hypothesise that our findings differ from those of (Mediratta et al., 2024) because they used `Expert` data, whereas we use mixed replay data. Expert data is more suitable for BC while many offline RL methods (especially CQL (Schweighofer et al., 2022)) benefit from having mixed data. Indeed, our LBF and Connector datasets are significantly more skewed towards `Expert` trajectories in the replay datasets because the tasks are easier than RWARE tasks. Hence why CQL likely did the best on RWARE, since those datasets are the most mixed. So in conclusion we find that in settings with mixed data quality offline MARL methods exhibit better generalisation than BC.

Table 2: *Comparison of test task performance of all three models.* The mean and standard error of the performance across all test tasks on `RWARE`, `LBF` and `Connector` for each of the multi-task algorithms (largest mean highlighted with bold). In the final column the combined mean across all tasks from the three environments is computed. **In contrast to the findings by Mediratta et al. (2024), we find that on each environment the best performing algorithm is an Offline RL method (MT CQL-Sable or MT Oryx), rather than the BC model. When aggregated across all the test tasks combined, MT Oryx performs the best.**

| | Algorithm | RWARE | LBF | Connector | Combined |
|---|---|---|---|---|---|
| ● | MT Oryx | $0.587 \pm 0.054$ | $\mathbf{0.803 \pm 0.026}$ | $\mathbf{0.852 \pm 0.002}$ | $\mathbf{0.759 \pm 0.023}$ |
| ● | MT CQL-Sable | $\mathbf{0.620 \pm 0.066}$ | $0.562 \pm 0.029$ | $0.668 \pm 0.018$ | $0.633 \pm 0.024$ |
| ● | MT BC-Sable | $0.415 \pm 0.050$ | $0.797 \pm 0.030$ | $0.775 \pm 0.004$ | $0.664 \pm 0.027$ |

# D  DATASET QUALITY ABLATION

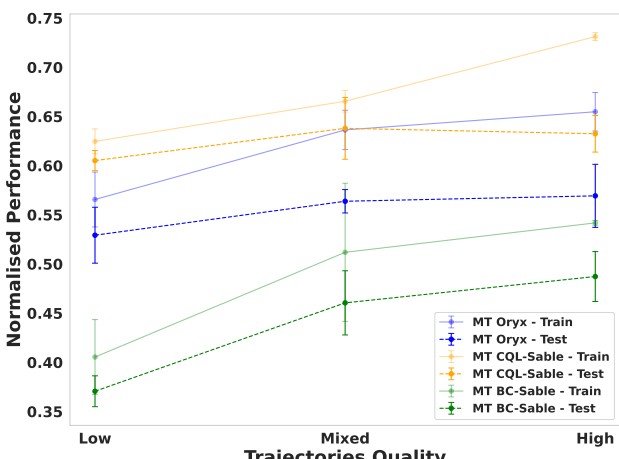

Figure 11: *Performance of MT-Oryx, MT-CQL-Sable, and MT-BC-Sable on RWARE with different trajectory subsets.* **High-quality trajectories improve training performance, particularly for MT-CQL-Sable, but these gains do not transfer to the test tasks. Low-quality trajectories consistently yield the worst results.**

**Do higher quality trajectories improve generalisation?** As observed in subsection 3.3, increasing dataset size does not lead to significant improvements in generalization to unseen tasks. A natural follow-up question is: how does the quality of trajectories in the dataset affect training and test performance? To investigate this, we conduct an experiment where training is performed with trajectories sampled from specific subsets of our dataset. Low-quality trajectories are those collected during the first two-thirds of the online training phase, while High-quality trajectories are those from the final third. Results on RWARE are shown in Figure 11. For all algorithms, training performance improves with High-quality trajectories, though the gains on test tasks remain marginal. Across all three algorithms, training with Low-quality trajectories consistently yields the worst results on both training and test tasks. These results suggest that the most effective strategy is to prioritize High-quality trajectories while retaining a small fraction of Low-quality ones as negative examples.

# E   SCALING ANALYSIS ON LBF AND CONNECTOR

In this section, we complement the experiments presented in subsection 3.3. We verify whether the model-size scaling trends observed in RWARE also extend to LBF and Connector. As shown in Figure 12, we observe similar behavior: performance improves with model size up to a critical point. However, both LBF and Connector are considerably easier than RWARE, and therefore their performance saturates at much smaller model sizes. Furthermore, although there is a large performance gap between BC-Sable and the other algorithms on LBF, the overall scaling trend remains visible, albeit more marginal.

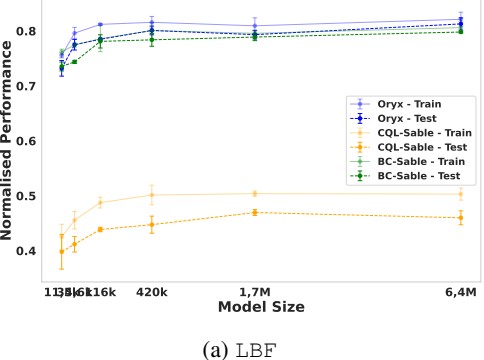
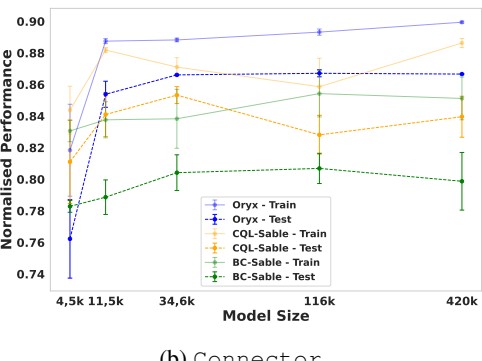

(a) LBF                                                        (b) Connector

Figure 12: *Performance of MT-Oryx, MT-CQL-Sable, and MT-BC-Sable on LBF and Connector with different model sizes.* **Both train and test performance of all algorithms improve with increasing model size up to a critical threshold, beyond which performance plateaus.**

# F  THE EFFECT OF THE TASK SPLIT ON SCALING TRENDS

**How does the train/test task split affect generalisation and performance scaling?** To answer this question, we repeat the model-size scaling experiment on `RWARE` using a different task split. Specifically, we adopt the split shown in Figure 13a. Unlike the previous split (see Figure 3), this configuration allows a clear decision boundary separating the train tasks from the test tasks. As a result, it reduces the potential for the learned strategies to interpolate across tasks. The results in Figure 13b confirm that the model-size scaling trends hold regardless of the task-split strategy. Nevertheless, this split yields a larger generalisation gap, as the model can no longer rely on interpolation to transfer strategies to unseen tasks.

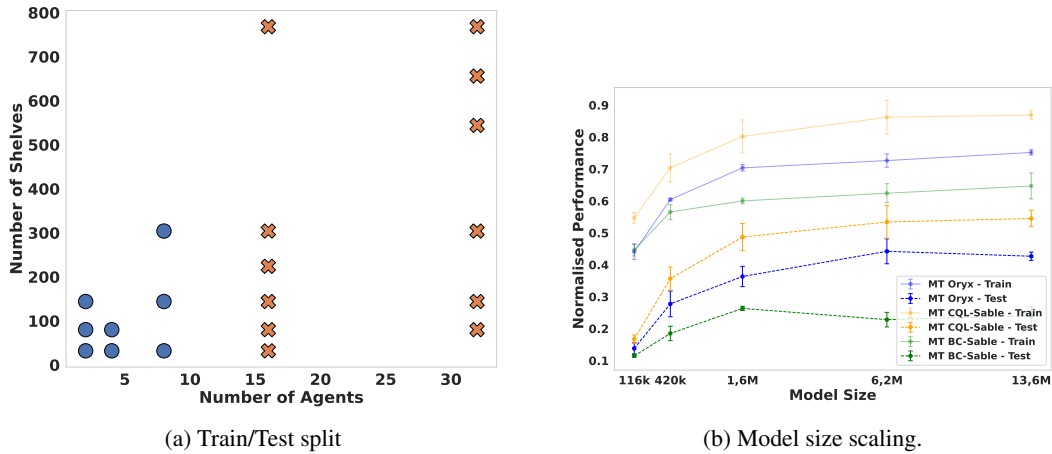

(a) Train/Test split  (b) Model size scaling.

Figure 13: Performance of MT-Oryx, MT-CQL-Sable, and MT-BC-Sable *(right) on* `RWARE` *environment with different model sizes using the train/test split on the (left)*. Similarly to Figure 5b we observe performance scaling with network size.

Finally, we conclude this analysis by repeating the task-scaling experiments using the new `RWARE` task split. The results in Figure 14 validate that the overall trends remain similar regardless of the split strategy. Test performance improves as the number of training tasks increases, while train performance decreases because it becomes more challenging for the model to learn a single strategy—or multiple strategies—that solve all tasks simultaneously.

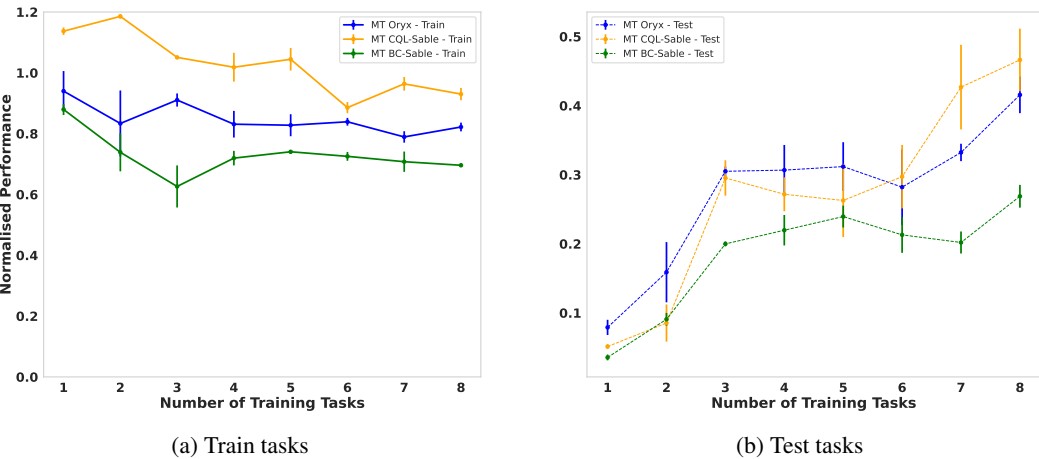

(a) Train tasks  (b) Test tasks

Figure 14: *Performance vs number of tasks with new* `RWARE` *task split.* We observe similar trends as in Figure 4. This results confirms that performance trend is independent of the task splitting strategy.

# G    FULL TRAINING CURVES

For additional insight into multi-task training dynamics we provide the complete set of training curves on RWARE and Connector. The plots are grouped by in-distribution (Training) tasks and out-of-distribution (Test) tasks.

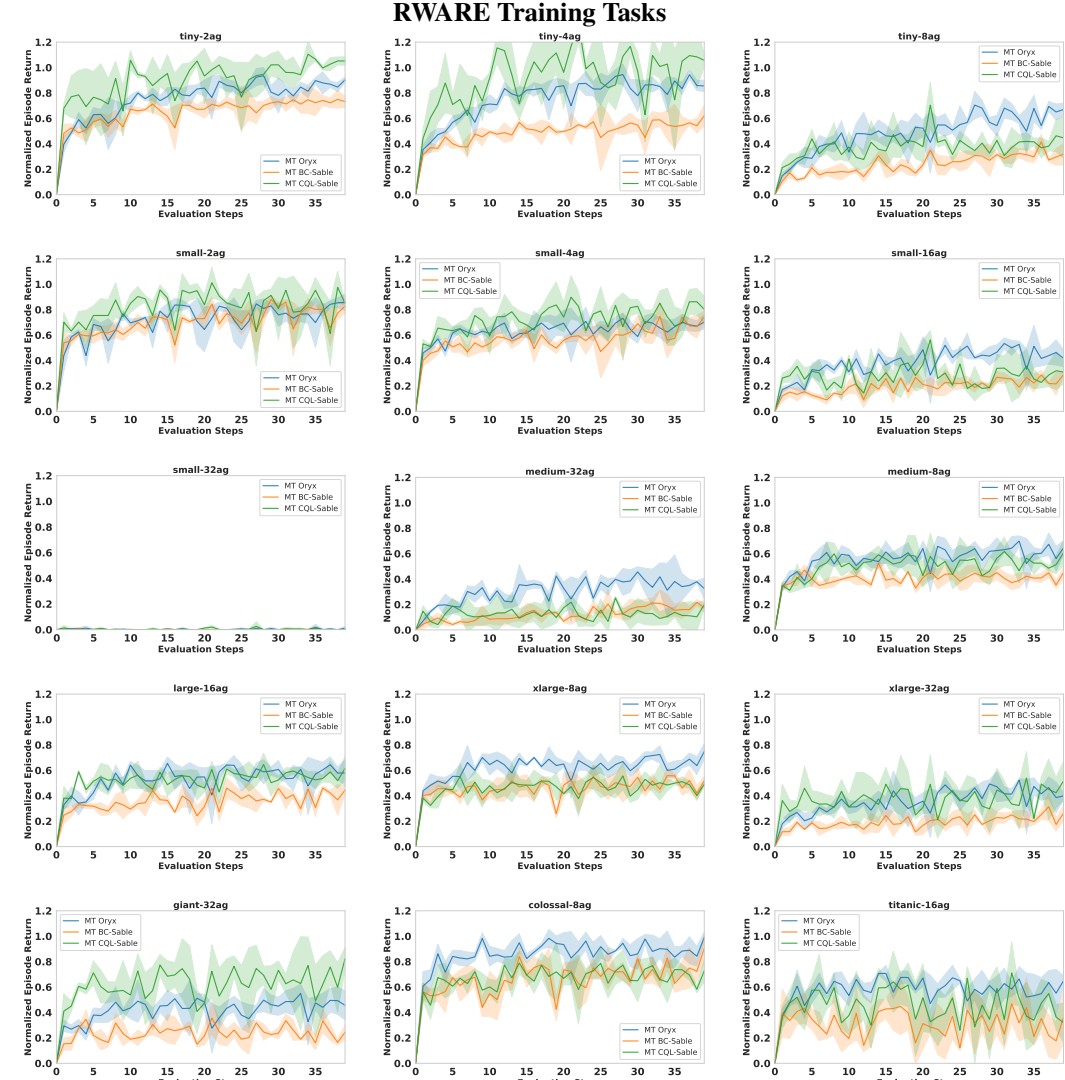

Figure 15: **In-Distribution (ID) Performance.** Evaluation curves for the 15 RWARE tasks where the agents were trained on distribution data.

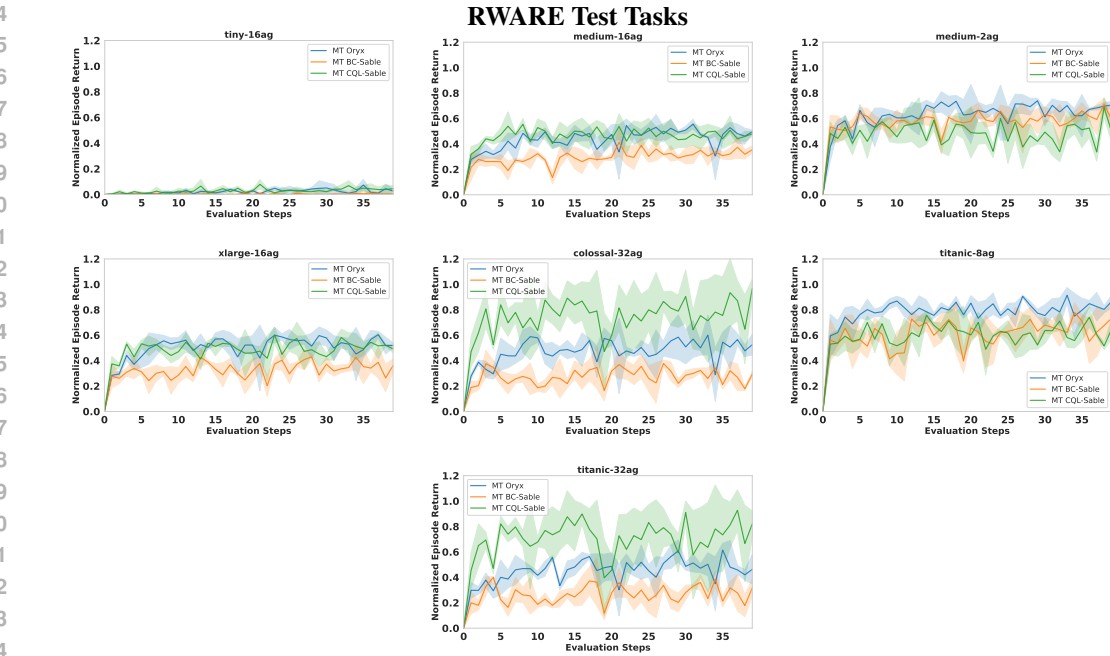

Figure 16: **Out-of-Distribution (OOD) Performance.** Evaluation curves for the 7 unseen RWARE scenarios to test generalization capabilities.

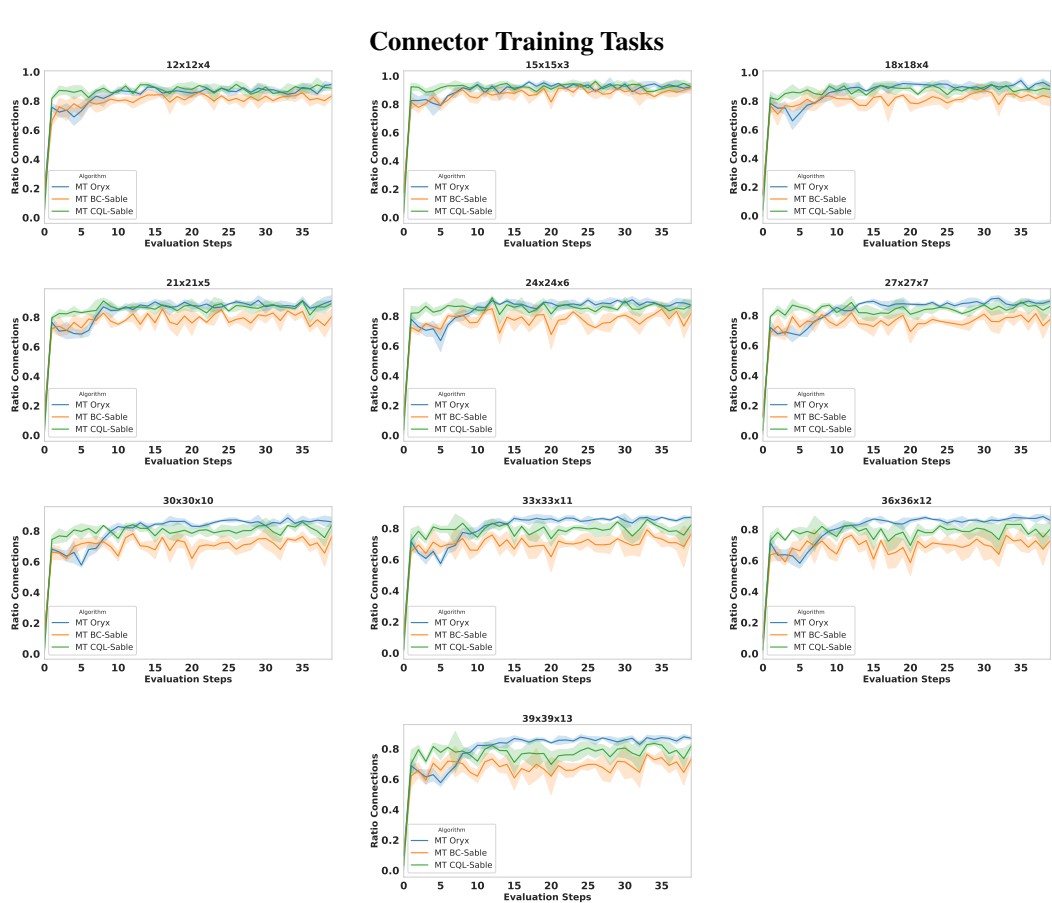

Figure 17: **In-Distribution (ID) Performance.** Evaluation curves for the 10 Connector tasks where the agents were trained on distribution data.

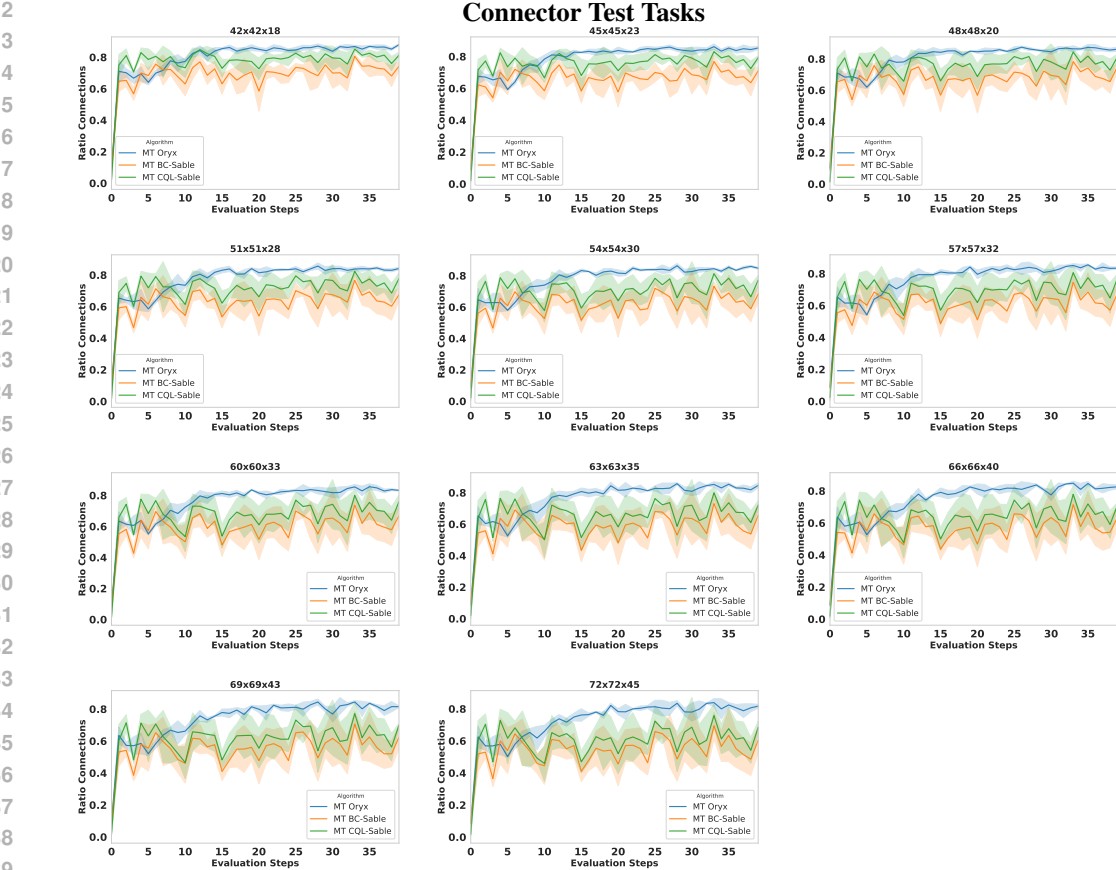

Figure 18: **Out-of-Distribution (OOD) Performance.** Evaluation curves for the 11 unseen Connector scenarios to test generalization capabilities.

# H   VISUALISATION OF MULTI-TASK POLICY

In order to qualitativly validate that the MT models have learn multiple team strategies which are quite distinct across tasks we visually inspected roll-outs across tasks. Here we visualise the learn strategy on two very distinct tasks `medium-2ag` and `medium-32ag`[2]. The main challenge in the first task is the sparsity of the warehouse. Accordingly the model learnt a strategy whereby the two agents rapidly traverse the warehouse to explore efficiently and find the shelf to be collected. In contrast, the central challenge on the second task is that the warehouse is very congested. If the agents collide the episode ends. Accordingly the model learnt a smart strategy of moving completed agents out of the way by sending them to the bottom right-hand corner. Importantly, a single MT model learn both of these different multi-agent strategies simultaneously.

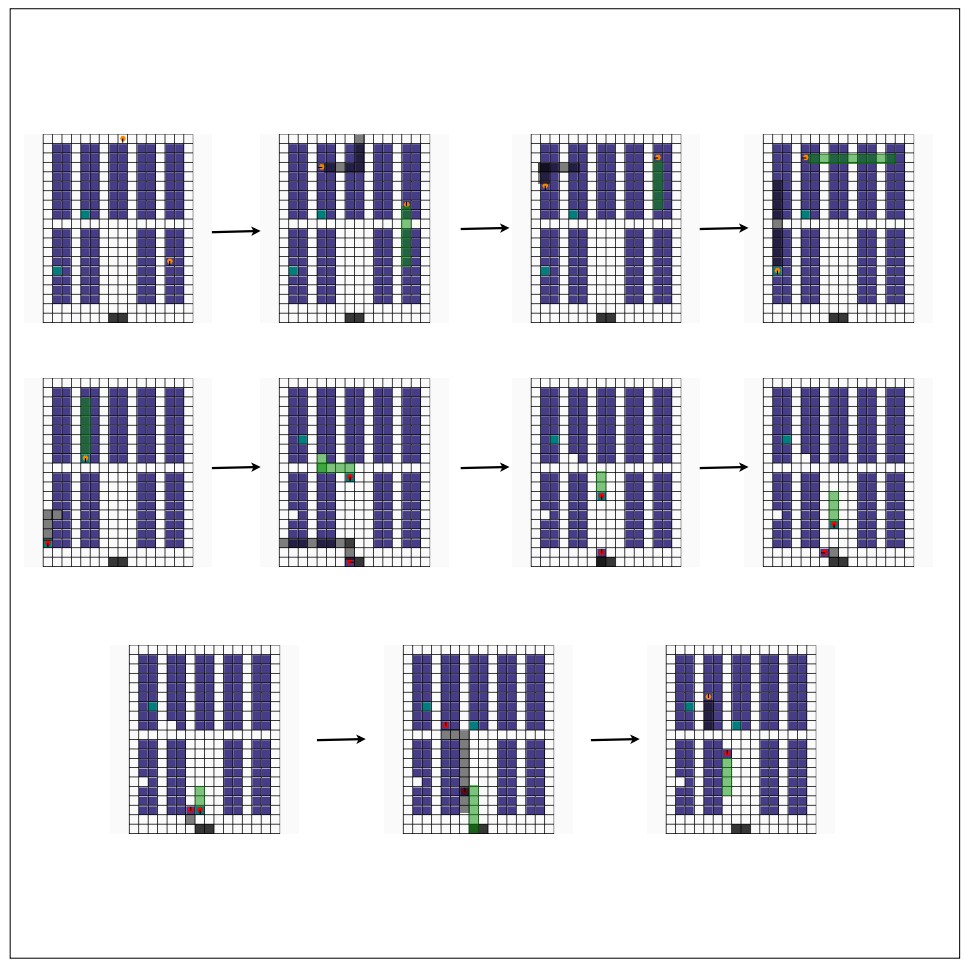

Figure 19: Visualisation of team strategy on `medium-2ag`. Frames should be read left to right, top to bottom.

---

[2]GIFs available on website: `https://sites.google.com/view/multi-task-marl`

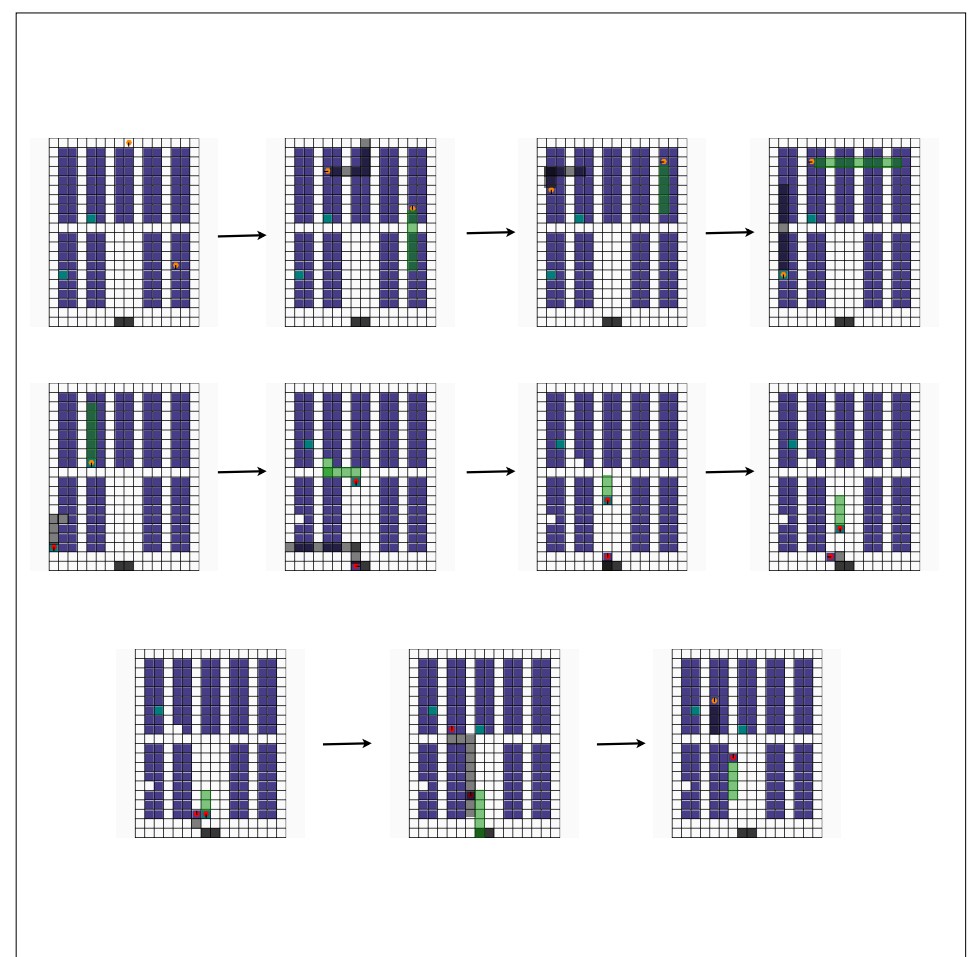

Figure 20: Visualisation of team strategy on `medium-32ag`. Frames should be read left to right, top to bottom.

## I    COMPUTATIONAL REQUIREMENTS

All experiments were conducted on a high-performance computing cluster utilizing the Jobset operator for orchestration. Each experimental run was allocated a single worker node equipped with one **NVIDIA A100-SXM4 GPU** (80 GB VRAM) and 24 logical cores of an **AMD EPYC 7742** processor.

The maximum wall-clock time for individual experiments was approximately 18 hours. We observed that computational resource usage remained consistent across all baselines, primarily because our setup avoids the use of task-specific heads. Furthermore, the retentive architecture inherent to the SABLE backbone—and by extension, Oryx—enables efficient scaling with respect to the number of agents. Consequently, our multi-task variants retain this computational efficiency even as environment complexity increases.

# J PRIMARY TASK SPLITS

To evaluate the generalization capabilities of our approach, we curated distinct sets of training and testing scenarios for each environment. The specific scenarios comprising each train/test split are detailed in Table 3.

Table 3: **Train/Test Task Splits for All Environments.** We list the specific scenarios used for training and out-of-distribution generalization testing.

| Environment | Split | # of Tasks | Scenarios |
|---|---|---|---|
| LBF | Train | 5 | {8x8-2p-2f, 10x10-3p-3f, 15x15-3p-5f, 15x15-4p-5f, 16x16-5p-6f} |
| | Test | 4 | {12x12-4p-5f, 14x14-3p-3f, 17x17-6p-8f, 17x17-8p-10f} |
| RWARE | Train | 15 | {tiny-2ag, tiny-4ag, tiny-8ag, small-2ag, small-4ag, small-16ag, small-32ag, medium-8ag, medium-32ag, large-16ag, xlarge-8ag, xlarge-32ag, giant-32ag, colossal-8ag, titanic-16ag} |
| | Test | 7 | {tiny-16ag, medium-2ag, medium-16ag, xlarge-16ag, colossal-32ag, titanic-8ag, titanic-32ag} |
| Connector | Train | 10 | {12x12x4a, 15x15x3a, 18x18x4a, 21x21x5a, 24x24x6a, 27x27x7a, 30x30x10a, 33x33x11a, 36x36x12a, 39x39x13a} |
| | Test | 11 | {42x42x18a, 45x45x23a, 48x48x20a, 51x51x28a, 54x54x30a, 57x57x32a, 60x60x33a, 63x63x35a, 66x66x40a, 69x69x43a, 72x72x45a} |

## K  HYPERPARAMETERS

This section details the hyperparameters used for our experiments.

Table 4: Default network settings for each environment.

| Parameter | LBF | Connector | RWARE |
|---|---|---|---|
| Model embedding dimension | 512 | 512 | 512 |
| Number of transformer heads | 4 | 4 | 4 |
| Number of transformer blocks | 1 | 1 | 1 |
| HL-Gauss value support | [-1, 1] | [-1, 1] | [-20, 20] |
| HL-Gauss number of bins | 51 | 51 | 51 |
| Sable's decay scaling factor | 0.8 | 0.8 | 0.8 |

Table 5: Default training settings.

| Hyperparameter | Value |
|---|---|
| Number of training updates | 60 000 |
| Number of evaluations | 600 |
| Number of evaluation episodes | 32 |
| Number of absolute evaluation episodes | 320 |
| Learning rate | $1 \times 10^{-3}$ |
| Discount ($\gamma$) | 0.99 |
| Polyak averaging coefficient ($\tau$) | 0.005 |
| Maximum gradient norm | 10 |
| Sample sequence length | 20 |
| Sample batch size | 480 |
| Value temperature | 1000 |
| Policy temperature | 0.1 |
| Critic loss coefficient | 1 |

Table 6: MT-Oryx specific settings.

| Hyperparameter | Value |
|---|---|
| Value temperature | 1000 |
| Policy temperature | 0.1 |
| Critic loss coefficient | 1 |
| HL-Gauss smoothing ratio | 0.75 |

Table 7: MT-CQL-Sable specific settings.

| Hyperparameter | Value |
|---|---|
| CQL loss coefficient | 10 |
| HL-Gauss smoothing ratio | 0.75 |

# L  DATASETS

## L.1  DATASET RELEASE PLAN

To guarantee the long-term reproducibility of this project, we will upload all of our datasets to a public HuggingFace repository[3]. This will be done upon publication of this work.

## L.2  DATASET STATISTICS

The following sections detail the statistics of the offline datasets for the RWARE, Connector, and LBF environments used in our experiments. Datasets were generated by recording rollouts from an online Sable (Mahjoub et al., 2025) agent at different intervals during its training. All data is collected from fixed intervals over training using an evaluation policy to vary the amount of data collected while maintaining a standard set of policies to sample from. For RWARE, we also create multiple datasets of different sizes by varying the number of evaluations sampled in order to perform our data-scaling experiments.

### L.2.1  RWARE

For our data-scaling experiments in the RWARE environment, we generated three offline datasets of varying sizes. The datasets were constructed by collecting 122, 244, and 610 evaluation rollouts from a pre-trained online Sable agent (Mahjoub et al., 2025). Table 8 provides detailed statistics for each dataset size across all RWARE scenarios, illustrating how the number of episodes and transitions scales with the number of collected rollouts.

### L.2.2  CONNECTOR

For the Connector environment, we generated 10 distinct offline datasets, one for each training scenario. Each dataset contains approximately 10 million transitions. The data collection process involved recording evaluation rollouts at 50 different checkpoints during the training of an online Sable agent. At each checkpoint, we generated 160 rollouts of 1280 timesteps each, resulting in a total of $50 \times 160 \times 1280 \approx 10.24$ million transitions per scenario. The ten scenarios used to create these datasets are listed in Table 10.

### L.2.3  LBF

For LBF we collected all the the training data from an online Sable run for each LBF scenario.

---

[3]https://sites.google.com/view/multi-task-marl

Table 8: **RWARE dataset statistics across different data collection checkpoints.** We report the total number of episodes and timesteps (transitions) for each scenario, corresponding to datasets created from 122, 244, and 610 evaluation rollouts.

| | 122 Rollouts | | 244 Rollouts | | 610 Rollouts | |
|---|---|---|---|---|---|---|
| **Scenario Name** | **Episodes** | **Timesteps** | **Episodes** | **Timesteps** | **Episodes** | **Timesteps** |
| tiny-2ag | 15,616 | 7,493,913 | 31,232 | 14,934,862 | 78,080 | 37,382,071 |
| small-2ag | 15,616 | 7,511,771 | 31,232 | 15,091,627 | 78,080 | 37,504,501 |
| tiny-4ag | 15,616 | 6,492,381 | 31,232 | 13,208,433 | 78,080 | 33,110,502 |
| small-4ag | 15,616 | 6,611,283 | 31,232 | 13,496,720 | 78,080 | 33,733,571 |
| tiny-8ag | 15,616 | 4,704,862 | 31,232 | 9,748,756 | 78,080 | 24,647,669 |
| medium-8ag | 15,616 | 2,502,476 | 31,232 | 5,148,947 | 78,080 | 12,747,091 |
| xlarge-8ag | 15,616 | 5,816,385 | 31,232 | 11,008,538 | 78,080 | 29,167,762 |
| colossal-8ag | 15,616 | 4,804,325 | 31,232 | 12,078,452 | 78,080 | 29,830,317 |
| small-16ag | 15,616 | 3,681,321 | 31,232 | 7,405,046 | 78,080 | 15,598,958 |
| large-16ag | 15,616 | 3,946,296 | 31,232 | 6,158,419 | 78,080 | 18,731,422 |
| titanic-16ag | 15,616 | 4,361,204 | 31,232 | 10,498,182 | 78,080 | 17,223,581 |
| small-32ag | 15,616 | 317,038 | 31,232 | 639,868 | 78,080 | 207,539 |
| medium-32ag | 15,616 | 4,147,400 | 31,232 | 8,386,685 | 78,080 | 20,855,336 |
| xlarge-32ag | 15,616 | 3,275,217 | 31,232 | 6,593,539 | 78,080 | 16,388,466 |
| giant-32ag | 15,616 | 3,682,013 | 31,232 | 6,513,235 | 78,080 | 12,706,872 |

Table 9: **Connector dataset statistics.** We generated a separate dataset of approximately 10.24 million transitions for each of the ten training scenarios.

| Scenario Name | Total Timesteps |
|---|---|
| 12x12x4a | $\approx 10.24 \times 10^6$ |
| 15x15x3a | $\approx 10.24 \times 10^6$ |
| 18x18x4a | $\approx 10.24 \times 10^6$ |
| 21x21x5a | $\approx 10.24 \times 10^6$ |
| 24x24x6a | $\approx 10.24 \times 10^6$ |
| 27x27x7a | $\approx 10.24 \times 10^6$ |
| 30x30x10a | $\approx 10.24 \times 10^6$ |
| 33x33x11a | $\approx 10.24 \times 10^6$ |
| 36x36x12a | $\approx 10.24 \times 10^6$ |
| 39x39x13a | $\approx 10.24 \times 10^6$ |

We observe that performance on the training tasks remains high across all environments, even as the number of tasks increases

Table 10: **LBF dataset statistics.** We generated a separate dataset of approximately 4 million transitions for each of the 5 training scenarios.

| Scenario Name | Total Timesteps |
|---|---|
| 8x8-2p-2f | $\approx 3.99 \times 10^6$ |
| 10x10-3p-3f | $\approx 3.99 \times 10^6$ |
| 15x15-3p-3f | $\approx 3.99 \times 10^6$ |
| 15x15-4p-5f | $\approx 3.99 \times 10^6$ |
| 16x16-5p-6f | $\approx 3.99 \times 10^6$ |

