# OpenReview forum: "Multi-Task Sequence Models Generalize in Offline Multi-Agent Reinforcement Learning"
_ICLR.cc/2026/Conference — Submitted to ICLR 2026_

### Official Review · Reviewer_7Eoh · 2025-10-28

**Soundness:** 2
**Presentation:** 3
**Contribution:** 2
**Rating:** 4
**Confidence:** 4

**Summary:**

The paper investigates generalization in offline multi-agent reinforcement learning (MARL) through multi-task training. The authors propose modifications to existing sequence models (introducing BC-Sable, CQL-Sable, and adapting Oryx) to handle multiple tasks with varying agent numbers. Key contributions include: (1) a multi-task offline MARL benchmark across three environments (LBF, RWARE, Connector), (2) demonstrating that multi-task training significantly improves zero-shot transfer to unseen tasks (219% average improvement), and (3) showing that dataset diversity matters more than dataset size for generalization.

**Strengths:**

- Well-structured and clearly written paper addressing the under-explored regime of offline MARL in multi-task settings.
-Strong empirical results with substantial performance gains (up to 442% on RWARE) demonstrating the effectiveness of multi-task training for zero-shot generalization.
- Comprehensive experimental setup with three different algorithms tested across three environments, showing consistent improvements from multi-task training.
- Important finding on dataset diversity vs. size: Clear experimental evidence (Section 3.4) that increasing task diversity improves generalization more than simply scaling dataset size.
- Thorough ablation studies (Section 3.5) validating each design choice, particularly showing 37% performance drop without task-balanced batching.
- Contradicts prior pessimistic findings by demonstrating that offline RL methods (particularly MT Oryx) can outperform behavior cloning, contrary to Mediratta et al. (2024).
- Multiple baselines and fair comparisons including two newly contributed baselines (BC-Sable and CQL-Sable) for this work.
Code and dataset availability with promised public release upon publication.

**Weaknesses:**

- Limited algorithmic novelty: The actual differences between BC-Sable, CQL-Sable and Oryx-based models are unclear. The paper primarily applies known techniques (task-balanced batching, HL-Gauss, agent masking) rather than introducing fundamentally new methods.
- Lack of theoretical justification: No theoretical analysis, proofs, or theorems explaining why intra-task transfer aids representation learning for generalization. The empirical results lack theoretical grounding.
- Insufficient task and environment context: The main text lacks adequate explanation of what agents do in each environment and why these tasks are challenging/beneficial for multi-task learning.
- Section 3.4 appears more exploratory than contributory: The findings about dataset/model scaling largely iterate on well-known ideas from function approximation and supervised learning. As noted, Mediratta et al. (2024) has similar results, raising questions about scientific contribution.

Inconsistent experimental design:
- Different numbers of training tasks across benchmarks (5 for LBF, 10 for Connector, 15 for RWARE) without clear justification
RWARE specifically chosen for scaling experiments without explanation
- Only best checkpoint results shown rather than average performance

- Many undefined abbreviations throughout the main text (Dec-POMDP, SABLE, RWARE, LBF) that reduce readability.
- Missing comparisons to other recent multi-task MARL methods mentioned in related work.

**Questions:**

- Line 048, Figure 1: What type of normalization is used for test performance? This is crucial for interpreting the 442% improvement claims.
- Line 120, Figure 2: Which specific task is shown in the visualization, and from which dataset?
- Line 130: What is a "Dec-POMDP"? This acronym needs definition when first introduced.
- Line 249: Why show only the best checkpoint? Is this common practice? Wouldn't average performance across checkpoints provide more robust evaluation of the approach?
- Line 259: Why use different numbers of training tasks for different benchmarks? Is this due to environment complexity, data availability, or other factors?
- Line 329: When "MT Oryx performs the best" aggregated across tasks, what specific properties allow it to leverage offline RL that BC struggles with? What makes this finding different from Mediratta et al. (2024)?
- Line 350, Experiment (b): Why was RWARE specifically chosen for model scaling experiments? Do other environments show similar trends? Also, why only vary embedding dimensions rather than exploring different scaling strategies for encoder vs. decoder?-
- General: How do your methods compare to other multi-task MARL approaches like MaskMA or HiSSD mentioned in related work? MADT are Decision Transformers and sequence models, like in your case.
- Reproducibility: What are the computational requirements (training time, memory) for the multi-task models compared to single-task variants?

---

> ### Author Response · Authors · 2025-11-24
> **Thanking the reviewer and addressing questions**
>
> Dear reviewer, thank you for taking the time to assess our work in detail.
>
> ## Rebuttal
>
> **Limited algorithmic novelty:**
> While the single task algorithms build on existing work, we would argue that contributing the multi-task versions of these algorithms should not be underestimated. Furthermore, we argue  that our work empirically demonstrates novel insights previously unexplored  in the multi-agent setting and that this is a valuable contribution to the research community.
>
> **Lack of theoretical justification:**
> We have added some additional analysis to the paper (see discussion in 3.2 and appendix). But we agree that working on theoretical justifications remains a valuable direction for future work.
>
> Further, in Appendix D we conduct a qualitative analysis of the learnt behavior by the model on two different RWARE tasks, one with only two agents and another with 32 agents. We show how the model simultaneously learnt two very different team strategies.
>
> **Inconsistent experimental design:**
> The different number of training tasks across environments reflects the relative difficulty of the environments we observed empirically. The benefit of adding more tasks to the training set far more rapidly saturates on LBF and Connector than on RWARE (as shown in Figure 4). Therefore, it did not make sense to us to add more tasks in LBF and Connector. Additional network scaling experiments have been added to Appendix B.
>
> **Insufficient environment context:**
> We apologize for this, we will add environment descriptions before the camera ready version is released.
>
> **Section 3.4 appears more exploratory than contributory:**
> Trends from Supervised Learning can not be taken for granted in RL [1]. More so in MARL, where we would argue that our work is one of the first to study this in depth. A key aspect of our contribution is the unprecedented scale of our experiments, which significantly moves the field forward: we varied the number of training tasks extensively to find the limits of generalization, unlike prior work that used a fixed, small set (3 training tasks in [2] and [3]). Furthermore, our multi-task models were evaluated simultaneously on 22 tasks in RWARE (compared to only 12 [2] and [3]) and scaled to tasks with up to 45 agents (compared to 12 in [2] and [3]). We therefore believe that there is immense value for the MARL research community in our exploration of the scaling trends.
>
> **Many undefined abbreviations:**
> We have now added definitions the first time we introduce an abbreviation throughout the text. We apologize for the confusion.
>
> **Missing comparisons:**
> For MaskMA [4]  we could not find any code online to work from. We also considered ODIS [2] and HiSSD [3], however, their code was overly specialised to SMAC (HiSSD claimed to have experiments on MAMuJoCo but code only supported SMAC). As a consequence we could not run their code on our environments and datasets. Nonetheless, we believe we still provided sufficient evidence with three different sequence models to substantiate our core claims about sequence model scaling trends in multi-task MARL settings.
>
> **Why show only the best checkpoint:**
> Using the best checkpoint during training is common in the MARL literature and related work [3]. However, we agree it may be useful for researchers to see the performance across training. So, we have added training curves across all tasks to Appendix C.
>
> **What type of normalization:**
> As described in section 3.1, in the paragraph titled “Evaluation protocol”.
>
> **Which specific task is shown in the visualization:**
> The tasks in the illustration are from RWARE. We apologize for the confusion, we have added this detail to the caption.
>
> **What makes this finding different:**
> Mediratta et al. used Expert data in their experiments, whereas our datasets are mixed. It is well known that many Offline RL methods (especially CQL) benefit from mixed data, whereas BC does best on expert data.
>
> **Why was RWARE specifically chosen for model scaling experiments:**
> We included RWARE scaling experiments in the main text because it was the most challenging setting. However, we did also perform the network scaling experiments on Connector and LBF and similar scaling trends hold, just far smaller network sizes. We have added these results to the appendix.
>
> **Computational analysis:**
> Thanks for pointing out that we did not include a computational analysis, this was an oversight on our part. We have now added one to the appendix.
>
> ## References
> [1] Kumar et al. Offline Q-Learning on Diverse Multi-Task Data Both Scales And Generalizes, 2022
>
> [2] Fuxiang Zhang, et al., Discovering Generalizable Multi-agent Coordination Skills from Multi-task Offline Data, 2023
>
> [3] Sicong Liu, et al., “LEARNING GENERALIZABLE SKILLS FROM OFFLINE MULTI-TASK DATA FOR MULTI-AGENT COOPERATION”, 2025,
>
> [4] Jie Liu, et al., MaskMA: Towards Zero-Shot Multi-Agent Decision Making with Mask-Based Collaborative Learning, 2024

---

> > ### Comment · Reviewer_7Eoh · 2025-11-25
> > **Concerns mainly emperically addressed**
> >
> > I thank the authors for their detailed response and the additional experiments added to the appendix. However, several core concerns remain insufficiently addressed:
> >
> > **1. Limited Algorithmic Novelty** The authors state that "the multi-task versions of these algorithms should not be underestimated." While I appreciate this perspective, a top-venue publication requires more than engineering extensions. The paper would benefit from conceptual or formal justification of _why_ and _how_ multi-task training aids representation learning—for instance, how task diversity shapes the learned manifold or induces beneficial inductive biases. Without such grounding, the contribution remains primarily empirical.
> >
> > **2. Lack of Theoretical Justification** The authors acknowledge this gap and defer theoretical analysis to "future work." While I appreciate the added qualitative visualization in Appendix D, this does not substitute for formal or even conceptual analysis. The observation that multi-task training changes agent cooperation dynamics (Appendix J) is interesting but remains descriptive rather than explanatory. For a work claiming novel insights about generalization in offline MARL, some theoretical grounding—even informal arguments connecting to representation learning literature—would significantly strengthen the contribution.
> >
> > **3. Insufficient Environment Context** The authors promise to add environment descriptions in the camera-ready version. This is concerning, as such fundamental context should be addressed during the discussion period to allow reviewers to properly assess the experimental setup.
> >
> > **4. Section 3.4 as Contribution** The authors argue that supervised learning trends cannot be assumed to hold in RL/MARL. While this is true, simply demonstrating that known scaling phenomena also appear in a new domain, without explaining the underlying mechanisms, raises more questions than it answers. The unprecedented scale of experiments is valuable for the community, but scale alone does not constitute a scientific contribution at the level expected of a top venue.
> >
> > **5. Missing Baselines** The practical difficulties in running MaskMA, ODIS, and HiSSD are understandable. However, this limits the paper's ability to position itself within the existing multi-task MARL literature. At minimum, a qualitative comparison discussing why these methods might or might not apply would strengthen the related work section.
> >
> > **In summary**, the authors have made commendable efforts to address presentation issues (abbreviations, training curves, computational analysis) and provide additional experiments. However, the fundamental limitations—lack of theoretical grounding, limited novelty beyond empirical scaling, and missing multi-task MARL baselines—remain. Given these concerns, I maintain my assessment, though I acknowledge its potential value as an empirical study and would not strongly oppose acceptance if other reviewers find the empirical contributions sufficient.

---

> > > ### Author Response · Authors · 2025-11-28
> > > **Theoretical underpinnings now added along with SMAC experiments**
> > >
> > > Dear reviewer, we thank you for your ongoing engagement with our work.
> > >
> > > **Insufficient Environment Context:** We sincerely apologize for this and agree with the reviewer that the additional environment context is essential for you to make an informed decision. We have now added details about the environments to the Appendix.
> > >
> > > **Limited Algorithmic Novelty & Lack of Theoretical Justification:** To further address your concerns we have added our theoretical underpinnings for improved generalization to section 2.2. We also provide an extensive explanation of our theoretical analysis in Appendix A. We trust that these theoretical underpinnings will reassure the reviewer of the value of our method. Coupled with our extensive empirical evidence we believe our paper has a very significant amount of value to add to the research community and therefore deserves to be published at a top-venue.
> > >
> > > **Missing Baselines:** While not central to our paper's core message, we do agree that comparing prior works would help situate the work within the wider context of the literature. To this end, we concatenated the SMAC-specific pre-processing that ODIS used to our method and evaluated it on their SMAC tasks. The results of which are now included in Table 1 of section 3.5. Further to this, we conducted a task scaling experiment using ODIS expert datasets from the three training tasks of the SMAC Marine-Hard task set, and found that similar scaling trends hold.
> > >
> > > **Conclusion:** We once again would like to thank the reviewer for their feedback on our work. By addressing the theoretical limitations of our work and the missing baselines, we believe we have substantially improved the quality of our contribution. Having addressed all the concerns raised we hope the reviewer will consider improving their score.

---

### Official Review · Reviewer_CUDK · 2025-10-30

**Soundness:** 3
**Presentation:** 4
**Contribution:** 3
**Rating:** 8
**Confidence:** 2

**Summary:**

This paper studies offline multi-task reinforcement leaning with sequence models. They demonstrate that scaling task diversity and the number of tasks leads to greater generalisation beyond scaling single task datasets.

**Strengths:**

This paper is very well written and provides sound insights and interesting discussions of results, making it an enjoyable read. It proposes logical claims backed up with evidence across settings. It provides details on the generalisation gap as well as additional experiments on model/data scaling. The paper uses current strong sequence models for baselines.

**Weaknesses:**

Whilst there is novelty in application for offline MARL with sequence models, it seems the primary takeaway from this paper is that increasing number of tasks and diversity of tasks improves generalization. This is not a particularly novel insight and has been the motivation of multi-task and meta-reinforcement learning since its inception. This paper could benefit from additional baselines, particularly those from different architectures such as those using decentralised actors to increase the rigor of its contribution. The scaling experiment whilst interesting, could benefit from seeing how data requirement scale with model parameters.

**Questions:**

Given its dramatic effect, would you consider task-balanced batching to be the primary enabling component for multi-task MARL?
How would you expect these results to transfer to other dominant MARL architectures? Do you hypothesize that the generalization gap between MT Oryx and MT BC-Sable would widen or narrow in a decentralised and CTDE algorithms?

---

> ### Author Response · Authors · 2025-11-24
> **Thanking the reviewer and responding to questions**
>
> We thank the reviewer for their positive review. Our ablation showed that task balancing helped, but we would not go so far as to say it is the primary enabling component. Regarding whether these findings will extend to other architectures, we are unsure because it is common in other MARL algorithms to use simple MLPs or RNNs, whose scaling capacity is less well understood compared to the sequence model architectures we used. However, we believe it’s still a valuable direction for future work.
>
> Kindest regards,
> Authors

---

### Official Review · Reviewer_mjub · 2025-10-31

**Soundness:** 3
**Presentation:** 3
**Contribution:** 2
**Rating:** 4
**Confidence:** 5

**Summary:**

This paper validates an conclusion: In the offline MARL domain, the current best-performing class of models, Multi-Task sequence models, exhibit favorable scaling properties in terms of zero-shot generalization capability with respect to task diversity. Specifically, increasing task diversity during training can significantly enhance the model's zero-shot generalization ability on unseen tasks. However, when task diversity is held constant, merely increasing the size of the dataset does not improve generalization capability.

**Strengths:**

- The paper is easy to understand, with a clear logical structure and few typos.
- Most of the conclusions drawn in the paper are supported by corresponding experiments.
- The conclusion proven in the paper, that "increasing task diversity can significantly improve the model's zero-shot generalization capability", is somewhat inspiring for future work in this field.

**Weaknesses:**

- Inappropriate metric: I think the metric used is not suitable. Since it is about success rate, the increase could simply be measured by $yourSR - baselineSR$. Using $\frac{yourSR - baselineSR}{baselineSR}$ is counterintuitive and may exaggerate the conclusion obtaining from the results.
- Are the size of dataset of multi-task and single-task the same? That is, are the numbers of episodes the same? Because the paper keeps emphasizing that increasing task diversity is more effective than increasing the size of the dataset, it is crucial to clarify whether the size of the dataset has been increased to the same level as in the multi-task scenario. Note that I am aware that Section 3.4 of the paper proves that simply increasing the size of the dataset does not help improve generalization capability. However, in Figure 1, multi-task and single-task still need datasets of the same size to rigorously prove your claim.
- Inappropriate Evaluation protocol: The evaluation is based on the best checkpoint achieved during training, which is inappropriate. If an algorithm has high variance, its best checkpoint may perform well. The paper should provide the asymptotic performance curve of the evaluation SR as training progresses.
- "We observe that performance on the training tasks remains high across all environments, even as the number of tasks increases. This indicates that the model can successfully learn across multiple tasks simultaneously." This statement is not correct. In fact, this statement only holds in Connector. The authors later also mentioned the performance drop in RWARE and LBF. Therefore, using this statement as the first sentence of this paragraph is very unrigorous.
- The paper draws several conclusions inconsistent with prior work but does not explain why the opposite conclusions were reached. For example, (1) "The experimental conclusion of this paper is that offline has better generalization capability than BC, while Mediratta et al. (2024) concluded that BC has better generalization capability than offline." Why did the opposite experimental results occur? The paper does not provide an analysis. I believe it is very important to provide a reasonable explanation for "reaching conclusions opposite to prior work," at least with some insight-level analysis or explanation. (2) "Notably, this result contrasts with the single-task setting reported by Formanek et al. (2025), where the optimal embedding dimension was just 64, underscoring the unique potential of multi-task data for enabling scale." Similarly, this conclusion opposite to prior work also needs to be analyzed.
- Why not conduct experimental comparisons on SMAC (or SMAX, i.e., SMAC in JAX)? On the one hand, SMAC is a very commonly used (as far as I know, the most commonly used) benchmark in the MARL field. On the other hand, there have been previous works on task-level generalization on SMAC, such as DT2GS [1], ODIS [2], and UPDeT [3].

[1] Tian et al. Decompose a Task into Generalizable Subtasks in Multi-Agent Reinforcement Learning. NeurIPS 2023

[2] Zhang et al. Discovering Generalizable Multi-agent Coordination Skills from Multi-task Offline Data. ICLR 2023

[3] Hu et al. UPDeT: Universal Multi-agent Reinforcement Learning via Policy Decoupling with Transformers. ICLR 2021

**Questions:**

See Weaknesses.

---

> ### Author Response · Authors · 2025-11-24
> **Thanking the reviewer for their assesment and addressing questions**
>
> We thank the reviewer for their assessment of our work and appreciate the remark that our work is “…somewhat inspiring for future work in this field.” However, we emphasize that our key contribution is the unprecedented scale of our experiments. Unlike prior work (Zhang et al., 2023; Liu et al., 2025) which relied on a fixed set of 3 training tasks and capped evaluation at 12 tasks/agents, we extensively varied training tasks to probe generalization limits. Specifically, we evaluated on 22 tasks in RWARE and scaled to 45 agents in Connector (32 in RWARE), significantly exceeding the 12-agent limit of prior baselines.
>
> ## Rebuttal
>
> **Inappropriate metric:**
> We appreciate this suggestion, however, we note that percentage increases are standard in reporting generalization improvements in the multi-task RL literature (Kumar et al., 2022a; Mediratta et al., 2024), particularly when comparing across environments with different reward scales. Normalised performance reporting also makes comparisons simpler across multiple environments with different reward scales like this present in this paper (see section 3.1).
>
> **Are the size of the dataset of multi-task and single-task the same?**
> Yes. In these experiments, the multi-task and single-task datasets have the same size. This design choice allows us to independently isolate the effects of dataset size and dataset diversity on performance.
>
> **Inappropriate Evaluation protocol:**
> We acknowledge that reporting only best checkpoint performance may not capture variance. We will add:
> Learning curves showing test performance over training: We have now included plots showing test and train task performance as a function of training steps (averaged over seeds) to demonstrate asymptotic behavior (Appendix C).
>
> **Final checkpoint performance:** In addition to the best checkpoint, we will report performance at the final training checkpoint to show that results are not driven by cherry-picking high-variance peaks.
> However, we should note that using the absolute metric is standard practice in offline RL evaluation (Kumar et al., 2020; Formanek et al., 2025) and reflects realistic deployment scenarios where practitioners select best checkpoints based on validation performance during training.
>
> **We observe that performance on the training tasks remains high:**
> We have updated the paper to note that the performance remains “relatively” high across the different environments, especially for MT-Oryx.
>
> **Why Our Offline RL Methods Outperform BC (vs. Mediratta et al. 2024)**
> * Dataset Composition Effects:  Our results do align with Mediratta et al. (Section 3.2; Appendix B): BC outperforms CQL on LBF (near-expert data), whereas offline RL excels on RWARE and Connector (mixed-quality). This confirms CQL’s known data sensitivities (Schweighofer et al., 2022). Notably, MT Oryx (using ICQ loss) performs well across all environments, suggesting the autoregressive ICQ objective is more robust than CQL for multi-agent multi-task settings across mixed and expert data.
> * Different Problem Settings: Mediratta et al. studied single-agent environments, while we focus on cooperative multi-agent coordination. Findings which apply in the single-agent setting should not be taken for granted in multi-agent settings.
>
> The relative performance of offline RL vs. BC is domain and dataset-dependent. Our results suggest that in cooperative multi-agent settings with appropriate algorithmic choices, offline RL can outperform BC. Based on this we believe that our work complements rather than contradicts prior works.
>
> **Why Multi-Task Training Benefits from Larger Models (vs. Formanek et al. 2025)**
>
> Representational Complexity: This is not a contradiction but a result of task diversity. Single-task training optimizes at dim 64, whereas multi-task training (5–15 tasks) requires capturing complex coordination and rewards, optimizing at dim 512 (6.2M params).
> Evidence & Alignment: Figure 5b shows performance improving with capacity, consistent with Kumar et al. (2022a) regarding diverse datasets. We demonstrate that multi-task data is the prerequisite for scaling to improve generalization in offline MARL.
>
> **Why not SMAC/SMAX?**
> * Heterogeneity: SMAC scenarios have significantly different observation/action spaces (e.g., 3 Marines vs. 5 Stalkers). This necessitates task-specific heads, preventing the evaluation of single-model zero-shot transfer.
> * Prior Work: While DT2GS, ODIS, and UPDeT explore different aspects of generalization requiring fine-tuning or task-specific configurations (e.g., skills/subtasks), our focus is strictly on zero-shot generalization using pure sequence models without task-specific tuning.
>
> ## References
> Zhang et al. Discovering Generalizable Multi-agent Coordination Skills from Multi-task Offline Data. ICLR 2023
>
> Sicong Liu, et al., “LEARNING GENERALIZABLE SKILLS FROM OFFLINE MULTI-TASK DATA FOR MULTI-AGENT COOPERATION”, 2025,

---

> > ### Comment · Reviewer_mjub · 2025-11-27
> >
> > **Inappropriate Metric**
> >
> > When using reward as a metric, it is indeed intuitive to use percentage increases due to the varying reward ranges across different environments/tasks. However, when using success rate as a metric, the success rate for any environment/task should be a number between 0 and 1. Therefore, using percentage increases actually makes the results less intuitive.
> >
> > **Why not SMAC/SMAX?**
> >
> > "Heterogeneity: SMAC scenarios ..., **preventing the evaluation of single-model zero-shot transfer**".
> >
> > This statement is factually incorrect. Even though the observation/action spaces of different tasks in SMAC are heterogeneous, DT2GS, ODIS, and UPDeT have all constructed model architectures that are generalizable across SMAC's multi-task scenarios. The core approach is to use population-invariant operators such as DeepSet, Attention, Gaussian Product, etc. Works utilizing similar structures include [1].
> >
> > "Prior Work: While DT2GS, ODIS, and UPDeT ... **without task-specific tuning**".
> >
> > 1. DT2GS, ODIS, and UPDeT have achieved generalization through different methods compared to this paper, making a comparison necessary. Particularly, ODIS belongs to the same offline MARL domain as the method presented in this paper.
> > 2. Figure 4 of DT2GS and Table 1 of ODIS both demonstrate that they can generalize to unseeen target tasks in a zero-shot manner. Therefore, task-specific tuning is not essential. Especially for ODIS, their main experiment (Table 1) explicitly states that they did not fine-tune on unseen target tasks but directly deployed their models.
> >
> > [1] Hao et al. Boosting Multiagent Reinforcement Learning via Permutation Invariant and Permutation Equivariant Networks. ICLR 2023
> >
> > I still believe that this paper needs to:
> >
> > 1. Conduct a more in-depth survey of task-level generalization work in the MARL domain;
> > 2. Perform experiments on more commonly used platforms in the MARL domain, such as SMAC, and compare with more baselines on it.

---

> > > ### Author Response · Authors · 2025-11-28
> > > **SMAC experiments and comparison to ODIS now added**
> > >
> > > We thank the reviewer for taking time to reevaluate our work and going through our responses.
> > >
> > > **Inappropriate metric:** We agree with the remark regarding success rates, however, the normalised performance we reported was not intended to be interpreted as a success rate. Having said that, we would like to avoid any potential confusion, and therefore propose now reporting relative improvements rather than percentage increases (See Line 52, Line 99 and Line 357). Please let us know if you are satisfied with this?
> > >
> > > **Why not SMAC/SMAX?:**
> > > We apologise, we never meant to imply that zero-shot transfer in SMAC is impossible, just that the varying action and observation spaces across tasks are prohibitive, requiring a complex layer of SMAC specific pre-processing. Moreover, even prior work had to keep the SMAC tasks they trained and tested on quite constrained in order for their SMAC specific pre-processing to work. For example, Table 1 of the ODIS paper included tasks with Marines only.
> > >
> > > Nevertheless, we agree that including SMAC, even in this limited form, would be beneficial. Thus, we leveraged the same pre-processor as ODIS and concatenated it to our models. We then compared our results with theirs. While not central to our paper's core message, we do agree it's valuable to see how our method compares to prior multi-task MARL algorithms. In addition, we conducted a task scaling experiment on SMAC using ODIS expert datasets from the three training tasks of the SMAC marine-hard task set, and found that similar scaling trends hold. All our results have been included in the main text under section 3.5.
> > >
> > > **Conclusion:** Having comprehensively addressed the reviewers concerns, we sincerely hope you will consider improving your score. Once again, thank you for your time and effort to review our work. It's been invaluable in strengthening our contribution.

---

### Official Review · Reviewer_FYMH · 2025-11-01

**Soundness:** 2
**Presentation:** 3
**Contribution:** 2
**Rating:** 2
**Confidence:** 4

**Summary:**

This paper investigates generalization in offline multi-agent reinforcement learning (MARL) using sequence models. The paper demonstrates that task diversity is more important than dataset size for achieving zero-shot transfer to unseen tasks. The paper introduces three multi-task sequence models (MT Oryx, MT CQL-Sable, and MT BC-Sable) and identify three key design choices for successful multi-task training: (1) task-balanced mini-batches, (2) value estimation as classification (HL-Gauss), and (3) agent masking/shuffling for variable team sizes. Experiments across three cooperative environments (LBF, RWARE, Connector) show that multi-task models achieve 219% average improvement on held-out test tasks compared to single-task models, and that offline MARL methods can outperform behavior cloning.

**Strengths:**

### Clarity
The paper is well written and easy to understand. The empirical findings are clearly presented and can provides actionable insights for practitioners, especially the insights that task diversity matters more than dataset size.

### Significance
The paper introduces three practical design contributions: task-balanced batching, HL-Gauss, agent masking. These identified design choices are simple, well-motivated, and effectively address multi-task MARL challenges. The ablation studies (Figure 6) validate their importance.

The observation that model capacity scaling improves generalization on difficult tasks (Figure 5b) is valuable and suggests promising directions for future work.

**Weaknesses:**

My major concerns are that the paper has limited novelty, and the empirical evaluation appears to be insufficient.

1. Limited novelty in the proposed models, and design choices:

**Incremental sequence models**: The paper's main algorithmic contributions are MT CQL-Sable and MT BC-Sable, which are essentially Sable (Mahjoub et al., 2025) with CQL and BC losses respectively. MT Oryx is Oryx (Formanek et al., 2025) adapted for multi-task settings. The core architectures (Sable's Retentive Network backbone, Oryx's ICQ formulation) are unchanged. The paper essentially shows these existing methods can be extended to multi-task settings with relatively minor modifications. Hence the claim that “We present two novel MARL sequence models (BC-Sable and CQL-Sable)” may not hold and the contributions appear very incremental.

**Limited technical depth in design choices**: The three design choices (task-balanced batching, HL-Gauss, agent masking) are sensible engineering decisions but not fundamental algorithmic innovations: Task-balanced batching is standard practice in multi-task learning (acknowledged via Cui et al., 2019 citation); HL-Gauss was already proposed by Farebrother et al. (2024) for handling varying reward scales; Agent masking/shuffling is a straightforward solution to variable team sizes


2. the empirical evaluation is insufficient and has limited insights.

**Insufficient analysis of generalization**: The paper demonstrates that multi-task training improves generalization, which has already been reported by many publications, e.g., [A generalist agent](https://arxiv.org/abs/2205.06175) and [Multi-Game Decision Transformer]( https://papers.neurips.cc/paper_files/paper/2022/file/b2cac94f82928a85055987d9fd44753f-Paper-Conference.pdf). The paper only provides limited insight into why or how. What shared structure are the models learning? Are certain task features more transferable? A representation analysis (e.g., visualization of learned features, attention patterns) would strengthen the work. Further, the discussion of task selection and the diversity measurement is limited: How were train/test splits designed to ensure meaningful distributional shift? What constitutes "diverse" tasks? Is it just varying parameters, or do tasks differ in structure? Would random task splits yield similar results, or is careful curation necessary?

**Missing multi-task MARL baselines**: the paper has no comparison with other multi-task MARL methods like MaskMA (which explicitly addresses multi-task MARL with varying agent/action spaces and shows strong zero-shot transfer on SMAC), ODIS (which tackles multi-task offline MARL via skill discovery) or HiSSD (which works on similar problem but uses hierarchical approach), though these are discussed in related work. How do the proposed methods compare to these specialized multi-task approaches?

**Questions:**

1. Figure 5b only shows results on RWARE with one algorithm. Do similar scaling trends hold for other environments and algorithms? The claim about scaling benefits needs broader support.

2. In Figure 6a the HL-Gauss ablation shows marginal benefits for MT CQL-Sable, which is a bit contradictory to the claim that it's essential for multi-task training. Can authors explain why HL-Gauss is not effective for MT CQL-Sable?

3. Some claims are overclaimed (e.g., "clearly show" in abstract when results are mixed). “a challenging multi-task ofﬂine MARL benchmark” but not any multi-task offline MARL methods have been benchmarked; “two novel MARL sequence models (BC-Sable and CQL-Sable)” clearly these are only incremental modifications of Sable.

4. Results may not be stable with only 3 seeds. Why not more seeds?

5. the paper only reports the normalized test performance. What about the computational cost: No discussion of training time, computational requirements, or efficiency. How practical are these methods for large-scale applications?

---

> ### Author Response · Authors · 2025-11-24
> **Thanking the reviewer for their helpful questions**
>
> Dear reviewer, thank you for taking the time to assess our work in detail.
>
> ## Rebuttal
> **Limited novelty in the proposed models:**
> We acknowledge that the single task models build upon existing methods. We will soften our claim about their novelty. However, we still contend that the extension to perform multi-task training is an important contribution.
>
> **Limited technical depth:**
> Our design choices were demonstrably valuable for improving performance, therefore, we included them in our write-up. However, we do not want these design choices to be misinterpreted as our work’s central contribution and will therefore soften the language we used in the text when referring to them.
>
> **Empirical evaluation insufficient:**
> It's true that in single-agent settings the question of multi-task generalisation has been explored, however, it should not be taken as given that these trends hold in the multi-agent setting. Our work is the first to empirically demonstrate multi-task and zero-shot generalisation scaling trends in MARL. Specifically, we varied the number of training tasks extensively to find the limits of generalization, unlike prior work that used a fixed, small set (3 training tasks in [1] and [2]). Furthermore, our multi-task models were evaluated simultaneously on 22 tasks in RWARE (compared to only 12 in [1] and [2]) and scaled to tasks with up to 45 agents (compared to 13 in [1] and [2]).
>
> Having said that, we have now added additional analysis of our results.
>
> In Appendix D we visualise the learnt behaviour by the model on two different RWARE tasks, one with only two agents and another with 32 agents. We show how the model simultaneously learnt two very different team strategies that are appropriate for the respective maps.
>
> Finally, in addition to the above, we added scaling experiments on Connector in Appendix A, experiments on the effect of task splits in Appendix B, full training curves in Appendix C (to shed light on training dynamics), and a compute analysis in Appendix E.
>
> **Designing diverse tasks:**
> We varied the number of agents across tasks and environment specific parameters (e.g. map layout). Trends do not rely on curated task splits. We have added a repeat of all the RWARE experiments using a different task split and we observe similar trends (Appendix B).
>
> **Missing multi-task MARL baselines:**
> For MaskMA [3] we could not find any open-sourced code online to work from. We also considered ODIS [1] and HiSSD [2], however, their code was overly specialised to SMAC (HiSSD claimed to have experiments on MAMuJoCo but code only supported SMAC). As a consequence we could not run their code on our environments and datasets. Nonetheless, we believe we still provided sufficient evidence with three different sequence models to substantiate our core claims about sequence model scaling trends in multi-task MARL settings.
>
> **Do similar scaling trends hold for other envs and algos?**
> Yes the scaling trends hold across all algorithms and other environments (Appendix B).
>
> **“HL-Gauss ablation shows marginal benefits for MT CQL-Sable…”**
> Since the performance improvement on MT Oryx was significant we thought it appropriate to report this design choice. Having said that, we agree the benefit should not be overstated.
>
> **“Some claims are overclaimed…”**
> We have now softened our claim about the novelty of the algorithms since we do not want to detract from the core contribution of this work, namely the empirical study into scaling multi-task learning in offline MARL for improved zero-shot generalisation.
>
> With regards to the use of the word “benchmark” we used that word here to mean a well defined set of experiments that can be run to evaluate algorithms. We don’t mean to say we “benchmarked” the field.
>
> **“Results may not be stable with only 3 seeds…”**
> Multi-task experiments are extremely computationally intensive. For comparison, [3] only used 4 seeds and [1] and [2] used 5 seeds. This despite not having the burden of sweeping across all subsets of training tasks ([1] and [2] had a fixed set of 3 training tasks). Nonetheless, we are busy running two extra seeds and will add them before the end of the rebuttal.
>
> **“The paper only reports the normalized test performance.”**
> We have now added to the appendix the full training curves across all tasks in our paper showing the training dynamics over time (Appendix C) . Further, we have added a computational analysis in the appendix (Appendix E) .
>
> ## References
>
> [1] Fuxiang Zhang, et al., Discovering Generalizable Multi-agent Coordination Skills from Multi-task Offline Data, 2023
>
> [2] Sicong Liu, et al., “LEARNING GENERALIZABLE SKILLS FROM OFFLINE MULTI-TASK DATA FOR MULTI-AGENT COOPERATION”, 2025,
>
> [3] Jie Liu, et al., MaskMA: Towards Zero-Shot Multi-Agent Decision Making with Mask-Based Collaborative Learning, 2024

---

> > ### Author Response · Authors · 2025-11-28
> > **Additional Results**
> >
> > Dear reviewer,
> >
> > We would kindly like to draw to your attention some recent changes we made to the paper. After constructive feedback from the other reviewers we have added theoretical underpinnings for our results in section 2.2. We expand in detail on the analysis in Appendix A. Further to this, we compared our method to ODIS using their multi-task SMAC test suit (Table 1, section 3.5). We also conducted a task scaling experiment on SMAC using their tasks. All additions to the paper are highlighted in green, we trust you will find them valuable.
> >
> > We believe that by comprehensively addressing the reviewers' concerns we have substantially improved our paper and hope you will consider improving your score. Once again, thank you for your time.

---

### Meta-Review · Area_Chair_QNTc · 2026-01-06

**Summary:**

This paper investigates the generalization capability of sequence models in offline MARL. The authors develops two strong baselines (MT BC-Sable and MT CQL-Sable), incorporating specific design choices (task-balanced batching, HL-Gauss value estimation, and agent masking) into training. Through experiments on LBF, RWARE, and Connector, the paper argues that increasing task diversity is more critical for zero-shot generalization than increasing dataset size alone.

While the reviewers appreciated the clarity of the writing and the extensive empirical results to support the paper's claim, the consensus leans towards rejection. The primary reasons for this assessment are the limited algorithmic novelty and incremental nature of the contribution.

**Reviewer Concerns:**

The authors made significant efforts during the rebuttal to address several concerns. In response to requests for traditional benchmarks and missing baselines, they added experiments on SMAC and comparisons to the ODIS algorithm. They also provided full training curves to clarify training dynamics and computational requirements. Furthermore, the authors attempted to address the lack of theoretical grounding by adding a section on "Theoretical Underpinnings" and providing qualitative visualizations of learned strategies.

However, fundamental concerns regarding novelty and the depth of contribution remain unresolved. Multiple reviewers maintained that the proposed models are essentially incremental combinations of existing tools. Despite the added theoretical section, reviewers found the analysis to be descriptive rather than explanatory, failing to rigorously justify how the specific multi-task training aids representation learning.
Finally, there remains disagreement on evaluation metrics, with one reviewer arguing that reporting percentage increases on success rates is potentially confusing.

**Reviewer Scores:**

Reviewer FYMH (2 -> 4): The review critically questioned the novelty and depth of the contribution, insufficient analysis and missing baselines. The authors partially addressed the concerns by adding explanations and experiments, however, the fundamental issue on the incremental nature of the contribution cannot be easily resolved.

Reviewer mjub (4 -> 4): This reviewer might have maintained the score. While the reviewer acknowledged the addition of SMAC experiments, the reviewer explicitly commented late in the discussion that he found the evaluation metrics inappropriate and felt the comparison with prior generalization works was insufficient.

Reviewer CUDK (8 -> 8): This reviewer was positive about the writing, methodology, and experimental results of the paper, and would likely to maintain the score.

Reviewer 7Eoh (4 -> 4): Despite acknowledging the authors' efforts, the reviewer is unsatisfied with the theoretical justification of the paper in the rebuttal.

---

### Decision · Program_Chairs · 2026-01-26

Reject